# The Initiation of a Phytosociological Study on Certain Types of Medicinal Plants

Emanuela Alice Luță [1,*], Manuela Ghica [2,*] and Cerasela Elena Gîrd [1]

1 Department of Pharmacognosy, Phytochemistry and Phytotherapy, Faculty of Pharmacy, University of Medicine and Pharmacy "Carol Davila", 6 Traian Vuia Street, 020956 Bucharest, Romania; cerasela.gird@umfcd.ro

2 Department of Applied Mathematics and Biostatistics, Faculty of Pharmacy, University of Medicine and Pharmacy "Carol Davila", 6 Traian Vuia Street, 020956 Bucharest, Romania

* Correspondence: emanuela.luta@drd.umfcd.ro (E.A.L.); manuela.ghica@umfcd.ro (M.G.)

**Abstract:** The cultivation of medicinal plants represents great necessity and topicality these days, given that the pharmaceutical industry requires high quality raw materials in large quantities. Those are used for the production of food supplements/phytomedicines/medical devices or gemmo-derivatives' products. Starting from these premises, this present study aimed to culture common batches of different associations of medicinal plants in order to quantify the fabrication of plant products, but also to observe possible changes in their internal structure, in direct correlation with the biosynthesis of active principles. The crops were monitored in 2018–2021. It was found that in all the common crops compared to the control ones, the amount of vegetable product provided was much higher (for example, the thyme-rosemary crop produced 730 g of fresh vegetable plants, compared with 540 g in the control crop; St. John's Wort in culture with lemon balm delivered 1934 g of vegetable product, compared with 1423 g obtained from the control crop; mint was grown with lemon balm and produced a double amount of vegetable mass compared with the control crop). The presence of numerous glandular hairs in the samples from the phytosociological groups for the species from the Lamiaceae family, could explain the difference in the volatile oil content (4 mL/100 g produced by rosemary from the thyme-rosemary crop compared with 3.6 mL/100 g from the control one; 6.6 mL/100 g generated by thyme from the thyme-rosemary crop compared with 3.6 mL/100 from the control group; 2 mL/100 g of lemon balm volatile oil from the mint-lemon balm compared with 0.6 mL/100 g). The content of other types of active principles is dependent on the culture association. From results analysis it was found that in the phytosociological groups, flavones, PCAs and total polyphenols were significantly higher compared to control ones ($2.4413 \pm 0.1858$ g flavones expressed in rutin/100 g in the thyme dried leaves from thyme-rosemary to $1.9317 \pm 0.0947$ g flavones produced by the control thyme; $9.9461 \pm 0.8385$ g PCAs expressed in chlorogenic acid/100 g for the same sample compared with $6.9709 \pm 1.4921$ g produced by the control batch; $11.1911 \pm 0.7959$ g TPC expressed in tannic acid/100 g in the thyme dried leaves from the thyme-rosemary phytosociological crop to $6.0393 \pm 0.3204$ g from the control one). The obtained results can be a starting point regarding the potential associations of medicinal plants in crops, in order to obtain a qualitative and quantitative vegetal mass.

**Keywords:** anatomical characteristics; phytosociology; raw vegetable materials; vegetation monitoring; active principles

---

## 1. Introduction

Science in the plant biology field is in continuous progress, and phytosociology, a branch of phytogeography, occupies an important place in research, in order to use plant communities as environmental indicators. By analogy with plant taxonomy, phytosociological classification (syntaxonomy) places vegetation units in a hierarchical system, based

on varying degrees of floristic resemblance [1]. Nowadays, this is the main method used throughout Europe and is also applied in North Asia and in different regions of Africa and Latin America [1]. While its application in North America has remained limited, recently it has launched the USNVC-US (National Vegetation Classification), which recognizes the importance of consistent hierarchical classification systems and adopts ideas from the Braun–Blanquet approach in a modified terminology [2]. While traditional phytosociologists believed that it can represent a whole set (portion) of vegetation with a single "typical" relevance, current phytosociology is seen as a statistical approach that aims to characterize vegetation types through combined information from several different plots [3]. The usefulness of molecular phylogeny and phylogenomics in speculating chemodiversity and bioprospecting is also highlighted within the context of natural-product-based drug discovery and development [4]. To understand the notion of a plant community, it has to be kept in mind that plant species that are components of a plant community grow together in a given area because they have similar requirements for existence [5] in terms of environmental factors, such as: light, temperature, humidity, nutrients, etc. [6]. Plant association can be defined as a group of plant species that grow together in a certain area and have a mutual alliance or affinity of a certain type between them. Following this relationship, it will be possible to observe, initially macroscopically and later microscopically, if this process will lead to changes or new acquired active principles [7]. The use of herbs in the treatment of certain diseases has been of high interest in recent decades. Different types of human communities traditionally use such herbs [8]. Some forest regions are extremely rich in their composition, and people then harvest them in excess, leading to the extinction of the species or even a change in the entire vegetation. A study was conducted on the natural medicinal plants in the Chalsa forest chain, located at the foot of the Himalayas, and sounded the alarm for a thorough phytosociological investigation, including an allelopathic analysis and an analysis of the soil seed bank [9]. In the study of Hatami E. et al. (2019), the importance of the symbiosis between plants and fungi was pointed out in the regeneration of soils contaminated involuntarily and/or voluntarily with oil residues [10]. Phytoremediation of lead-contaminated soils presented in the study organized by Saleem M et al. (2018), talks about the association of rhizobacteria in order to increase the resistance of plants to these types of contaminated soils [11].

A study conducted on five natural populations of *Terminalia chebula* Retz. realized by Singh S. et colb. (2019) aimed to analyze the structure of the vegetation and the distribution pattern of different species of trees and shrubs in these populations. It was able to provide information on the abundance, distribution, and rate of change in species composition [12]. Maharani D. et al. (2022) found that it is possible to optimize the land in the forest subsoil with the help of tubers that are resistant to shade, which can provide medium-term benefit for both forestry and food for the local population [13].

Plants that grow together have a common relationship with each other and with nature. The number of species sheltered in a community is an important factor from the ecological point of view since it seems to increase as the community becomes more stable. Those species accommodated in a plantation area were found in a higher sum compared with a degraded area [14]. According to studies by Jamil N. et al. (2022) and Koskey G. et al. (2022), one of the ways to increase the sustainability of some crops would be the reintroduction of intercropping crops; one of the main factors influencing the development of plants in the crop is the soil and climatic conditions, which can be translated into the horizontal or vertical evolution of those crops [15,16].

The interactions between certain types of plants and the environment have led to different types of vegetation occurring in different areas. The forests in Kandi Siwaliks, India, have witnessed the reduction of forested areas, isolation of smaller patches, habitat loss and a rise in disturbance level over the years. Therefore, to conserve plant diversity around the forest–village/town interfaces a protective buffer of edge species around newly created fragmented forest patches is required to protect the core species. The abundance to frequency ratio indicated that most of the species of shrub-sapling, herb-seedling and

trees were contagiously distributed except a few species of trees, which showed random distribution pattern. The massive deforestation found in different parts of the globe has shown a much lower floristic development of plant species that coexisted with different species of trees. The regeneration of the soils, but also of the green space specific to the forests, is dependent on the coexistence in the natural environment of these plant–tree associations, a fact observed in the subtropical forests, especially in the Asian area [17–19]. The study conducted by Li L. et al. (2022) on the intercropping of different plant species aimed at increasing the amount of biomass produced and as much as possible reducing the supplementation of soil with organic fertilizers. The adaptation of crops in intercalated zones as well as the production of a large amount of plant biomass can be explained by the significant increase in soil microbial diversity in these types of cultures, compared to their reduction in chemical crops [20].

The aim of this study is to generate compatible batches of medicinal plants that can be grown together in order to produce much higher plant mass, while also assuming that the content of secondary metabolites may be higher. Based on this premise, the objectives of the study were: the establishment of medicinal plant cultures in common crops and observation of their development over a longer period of time along with comparation to control crops. The plant mass production and the content of secondary metabolites (volatile oils, flavones, PCAs, and total polyphenols content) were also analyzed. In these plants, the medicinal association type of family and/or the therapeutic component were taken into account.

## 2. Materials and Methods

In this phytosociological study vegetable raw material with similar active principles have been selected, but belonging to different species or families. In order to carry out the scientific research, it was necessary to draw up a structured plan in stages conducted over several years, including both theoretical and practical parts. The following types of medicinal plants were associated in cultures based on therapeutic effects: crop 1—*Mentha × piperita* L. and *Melissa officinalis* L. (aromatic medicinal plants from the Lamiaceae family); crop 2—*Thymus vulgaris* L. and *Calendula officinalis* L. (associated in gastrointestinal diseases); crop 3—*Rosmarinus officinalis* L. and *Matricaria chamomilla* L. (source of volatile oil); crop 4—*Hypericum perforatum* L. and *Chelidonium majus* L. (associated in hepatobiliary disorders). Each phytosociological crop was accompanied by the corresponding control crop.

### 2.1. Description of the Study Site and Cultivation Condition

All of the medicinal plants were pre-planted in experimental crops with the size of 50 cm × 300 cm, and 400 cm intervals between batches. The distance between the planted seedlings was established depending on the height of the young plants that were planted >30 cm. Transplanting was performed at a depth of 25 cm and 5 seedlings/batch were used. The scientific experiment was carried out in the suburban area of Turnu Măgurele, Teleorman County, Romania (43°44′44.16″ N, 24°52′53.40″ E), starting in 2018. The average annual temperature in Turnu Măgurele is 11.5 °C, the average in the warm months is 23 °C, and the average in the cold months drops below −2 °C. It is characterized by a high caloric potential, high amplitudes of air temperature, low amounts of precipitation and often in a torrential regime in summer, and frequent periods of drought [21].

The crops were observed in 2018, and were then analyzed in the following years, including 2021. A series of techno-agricultural works were carried out (land preparation-shredding, leveling and irrigation, weed control, seedling production, replanting), as seen in Figure 1.

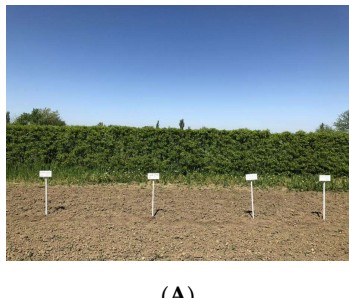
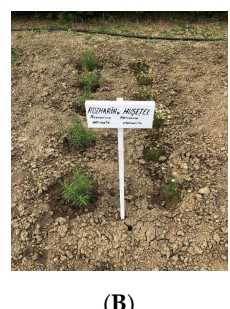
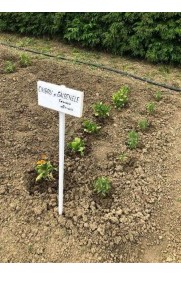

(**A**)  (**B**)  (**C**)

**Figure 1.** Developing the ground in two steps-place arrangement, 2018 (**A**), and planting the seedlings (**B**,**C**).

*2.2. Experimental Procedures*

The phytosociological crops as well as the control ones were monitored during their development (2018–2021) by the horticultural engineer Ciocăneală Ștefan, being subjected to the same operations such as watering, weeding, etc. Seedlings were dug from chemically untreated seeds, using prolonged germination to improve the quality of the active ingredients [22]. The replanting of seedlings with new medicinal plants, as well as the completion of existing ones, was done every year in April. The control crops were planted at a distance from the culture, in order not to be influenced. Each medicinal plant from the phytosociological crop had a control one consisting of the same number of seedlings.

During the maturation of the harvest, the morphological and phytochemical aspects were examined by comparing every crop with the control one. The research also sought to highlight potential morphological and anatomical changes in freshly harvested plant products using a stereomicroscope (ZEISS Stemi 508 Greenough Stereo Microscope with 8:1 Zoom, details up to 50× magnification).

*2.3. Dosing Active Chemical Constituents from Raw Materials Used in the Study*

The flavones, PACs (phenolcarboxylic acids) and total phenolic content were determined using spectrophotometric methods, as follows:

Preparation of the extractive solutions: 1.0000 g of vegetable products are refluxed with 50 mL of 50/70% ethanol (in order to establish the optimal concentration of solubilized active principles in the extraction solvent) for 30 min. The extractive solution is obtained by filtration in a 50 mL flask. It's brought to the mark with the same solvent. Those solutions are used for all types of determinations.

2.3.1. Dosing the Flavonoids Content (FL)

The flavonosides and their aglycones form, in the presence of aluminum chloride (neutral medium) and sodium acetate, products of reaction (chelates) with a yellow coloration and more intense fluorescence than the initial compounds.

Reagents and solvents: aqueous solutions of sodium acetate 100 g/L and aluminum chloride 25 g/L, rutoside (purchased from Sigma-Aldrich, Schnelldorf, Germany).

Carrying out the determination: volumes between 0.3–1.2 mL of extractive solution are placed in a 10 mL volumetric flask, treated with 2 mL of sodium acetate 100 g/L solution and 1 mL of aluminum chloride 25 g/L solution, then it is completed up to the mark with extractive ethanol. In parallel, a control sample is prepared, containing 1 mL of extractive solution and ethanol up to 10 mL. The absorbance of the sample is measured in comparation with the control one at the wavelength λ = 427 nm (maximum determined for metal ion chelation), on a 2005 Jasco V-530 spectrophotometer [23].

To determine the flavones' content, the previously constructed standard curve is used, using the following calculation formula:

$$c\% = \frac{Ep}{Eet} \times \frac{Cet}{Cp} \times 100$$

where:

$c\%$ = the flavonoid concentration of the sample (μg rutozid/100 g dried vegetable product);
$E_p$ = sample absorbance;
$E_{et}$ = the absorbance of the rutoside solution of a known concentration from the standard curve;
$C_{et}$ = the concentration of the rutoside solution corresponding to the measured absorbance (μg/mL);
$C_p$ = mass of dry vegetable product corresponding to 1 mL of sample solution used for dosing (g/mL).

A standard rutoside curve was previously established (Scheme 1). The technique used is that described for plant products, replacing the extractive solution with that of rutoside.

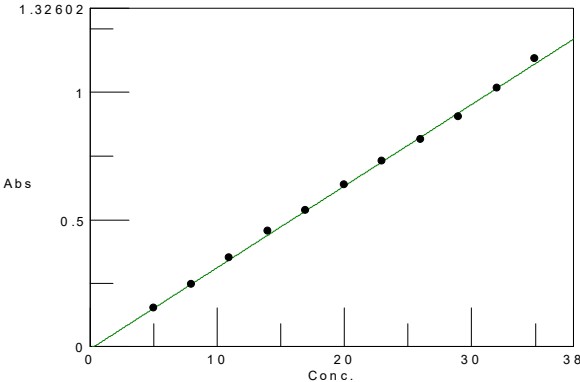

**Scheme 1.** Rutoside curve.

### 2.3.2. Dosing PAC (Phenolic Acids)

Phenolic acids react with nitric acid (released after the reaction between sodium nitrite and hydrochloric acid) to form nitroso derivatives, which spontaneously isomerize to isonitroso derivatives. Isonitroso derivatives, acidic compounds, are solubilized in an alkaline medium with the formation of red oxime.

Reagents and solvents: aqueous solutions of Arnow's reagent (sodium nitrite 100 g/L), hydrochloric acid 0.5 N, sodium hydroxide 85 g/L, chlorogenic acid (purchased from Sigma-Aldrich, Schnelldorf, Germany).

Carrying out the determination: volumes of extractive solution between 0.2–1 mL are successively treated with 2 mL hydrochloric acid 0.5 M, 2 mL Arnow's reagent and 2 mL sodium hydroxide 85 g/L, then it is brought up to the mark with distilled water. In parallel, a control is prepared, with a content similar to the sample, except for the Arnow's reagent. The absorbance of the sample is measured in comparation with a control at the wavelength $\lambda$ = 525 nm (maximum determined for chlorogenic acid oxime), on a 2005 Jasco V-530 spectrophotometer [23].

The determination of the total phenol-carboxylic derivatives was determined by interpolation on the chlorogenic acid standard curve. The same calculation formula as the previous type of active principles was also used here.

A standard chlorogenic acid curve was previously determined (Scheme 2).

### 2.3.3. Dosing the Total Phenolic Content (TPC)

The phenolic groups reduce the molybdenum derivatives $Mo^{6+}$ (colored in yellow) to molybdenum $Mo^{4+}$ and $Mo^{5+}$ (colored in blue), whose intensity is directly proportional to the concentration.

Reagents and solvents: Phosphotungstic acid (PTA), Folin–Ciocâlteu reagent, sodium carbonate 150 g/L, sodium carbonate 200 g/L, and tannic acid (purchased from Sigma-Aldrich, Schnelldorf, Germany).

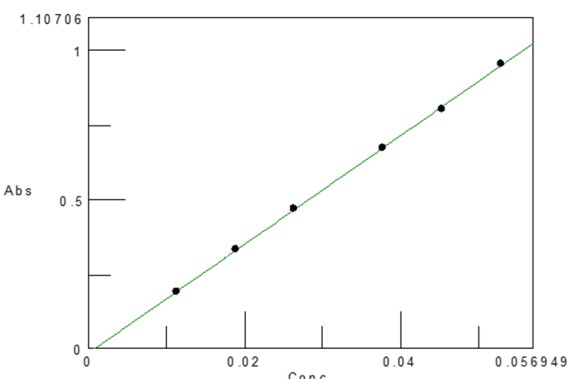

**Scheme 2.** Chlorogenic acid curve.

Performing the determination: volumes between 0.1–0.9 mL are diluted to 1 mL with water, treated with 1 mL of Folin–Ciocâlteu reagent and 8 mL of sodium carbonate 200 g/L. The control sample is obtained by replacing the extractive solution with water, adding 1 mL of Folin–Ciocâlteu reagent and 8 mL of sodium carbonate solution 200 g/L. The samples are incubated in the dark for 40 min. Subsequently, the absorbances at $\lambda = 725$ nm (the maximum determined for tannic acid), compared to the control samples, on a 2005 Jasco V-530 spectrophotometer [23]. To determine the total phenolic content, the same calculation formula was used, as above.

A standard of total phenolic curve was previously determined (Scheme 3).

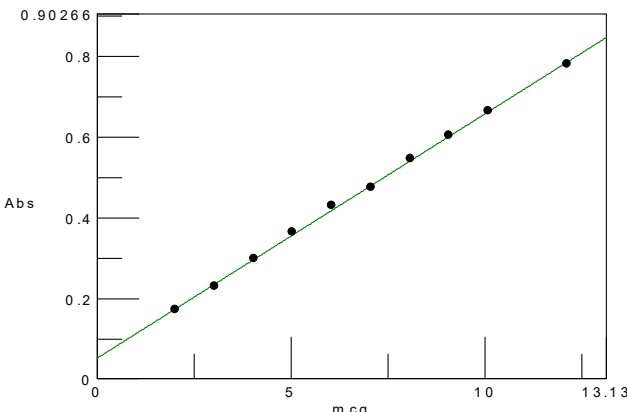

**Scheme 3.** Total phenolic curve.

### 2.4. Dosing Volatile Oil

Principle of the method: the property of volatile oil to be distillable with water vapor.

Steam distillation can be performed in a regular distillation apparatus (open circuit) or in a NeoClevenger type apparatus (closed circuit).

Working technique: 50 g of vegetable product and 500–1000 mL of solvent (usually water) are placed in an extraction flask adapted to the apparatus. The mixture thus obtained is subjected to heating directly in the flame, on a sieve. By reaching the boiling point of the extraction solvent, the volatile oil is distilled and captured in the graduated tube of the apparatus.

### 2.5. Statistical Analysis

Statistical analysis was implemented using the open-source software R (R version 4.1.1.) [24]. Our study contains data sets with small and unequal samples and this fact contradicts with the classical manner so, we use a similar parametric framework but with a bootstrap approach, without concerns about basic violation about normality, homoscedasticity and sphericity [25]. For simultaneous evaluation, the effect of two factors: Compound

(with three levels: FL, PCAss and TPC) and the Sample (with two levels: common crop and control crop) on a response variable named Concentration, we used a two-way robust ANOVA test for every Plant product [26]. Statistical significance is set to 0.05 (5%) and for post hoc analysis a Bonferroni-adjusted alpha level.

## 3. Results

*Plant Material*

The best results were obtained in the case of a common crop cultivation of two medicinal plants with a wide therapeutic use—*Mentha* × *piperita* L. and *Melissa officinalis* L. (Figure 2). At the time of introduction into the culture, the seedlings did not exceed 10 cm, but over the years they have developed and reached heights of 57 cm for mint and 89 cm for lemon balm [23].

Following the comparison of the phytosociological crop with the control one, the differences are obvious in terms of vertical and horizontal development dynamics for both species. Drawing a parallel between the crops in May and June 2018 with those in 2019, an abundance in the evolution of plants grown in the same crop, and especially for lemon balm, was observed [24]. We correlated this fact with an increase of precipitation (215 mm in 2019 compared to 209 mm in 2018), a maintaining of relative humidity at the same average values → 72, and a decrease of 1.3 degrees of maximum temperature recorded (Table 1). For the 2020–2021, the phytosociological crop has taken a different direction, with an increase in density for mint (Figures 2f and 3h).

From the point of view of an organoleptic analysis, it was possible to observe during 2018–2020 a significant development and growth of lemon balm in the phytosociological batch (Figure 2h), but in 2021 there is a visible difference, with the extension of mint to the detriment of lemon balm (Figure 3h). That result it can be associate with a temperature difference (maximum wave → 36.8 °C in 2021 compared to 33.1 °C in 2019), or of lower climatic conditions in precipitation compared to previous years (135 mm in 2021 compared to 215 in 2019) (Tables 1 and 2).

The horizontal evolution of the phytosociological crop mint-lemon balm for 2018–2021 led not only to an increase in the mass of plant product, but also to an overlap between species (Figure 3g,h). For a better evaluation, Figure 4 shows the evolutionary stages of *Mentha xpiperita* L. and *Melissa officinalis* L. growth in common crops compared with control ones during 2018–2021.

**Table 1.** Climatic conditions in Turnu Magurele town (Romania) during 2018–2019 according to ANM (National Weather Service) [27].

| Turnu Magurele | Period | Medium Value | Minim Value (Date) | Maxim Value (Date) | Number of Observations |
|---|---|---|---|---|---|
| T air (°C) at altitudes of 2 m above the ground | 01.05–30.06.2018 | +20.9 | +8.8 (13.05.2018) | +34.4 (13.06.2018) | 1452 |
| | 01.05–30.06.2019 | +20.1 | +5.6 (09.05.2019) | +33.1 (23.06.2019) | 1461 |
| P0, atmospheric pressure at the station level (mmHg) | 01.05–30.06.2018 | 756.9 | 747.5 (30.06.2018) | 764.0 (28.05.2018) | 1452 |
| | 01.05–30.06.2019 | 757.3 | 747.8 (05.05.2019) | 765.5 (26.06.2019) | 1461 |
| U, relative humidity (%), 2 m above the ground | 01.05–30.06.2018 | 72 | 24 (29.05.2018) | | 1452 |
| | 01.05–30.06.2019 | 72 | 25 (03.05.2019) | | 1461 |
| | | The amount of precipitation | Maxim Value (date) | The proportion of days with precipitation | Number of observations |
| RRR, the amount of precipitation (milimeters) | 01.05–30.06.2018 | 209 | 50.0 in 12 h (28.06.2018) | 31 | 121 |
| | 01.05–30.06.2019 | 215 | 42.0 in 12 h (25.06.2019) | 24 | 122 |

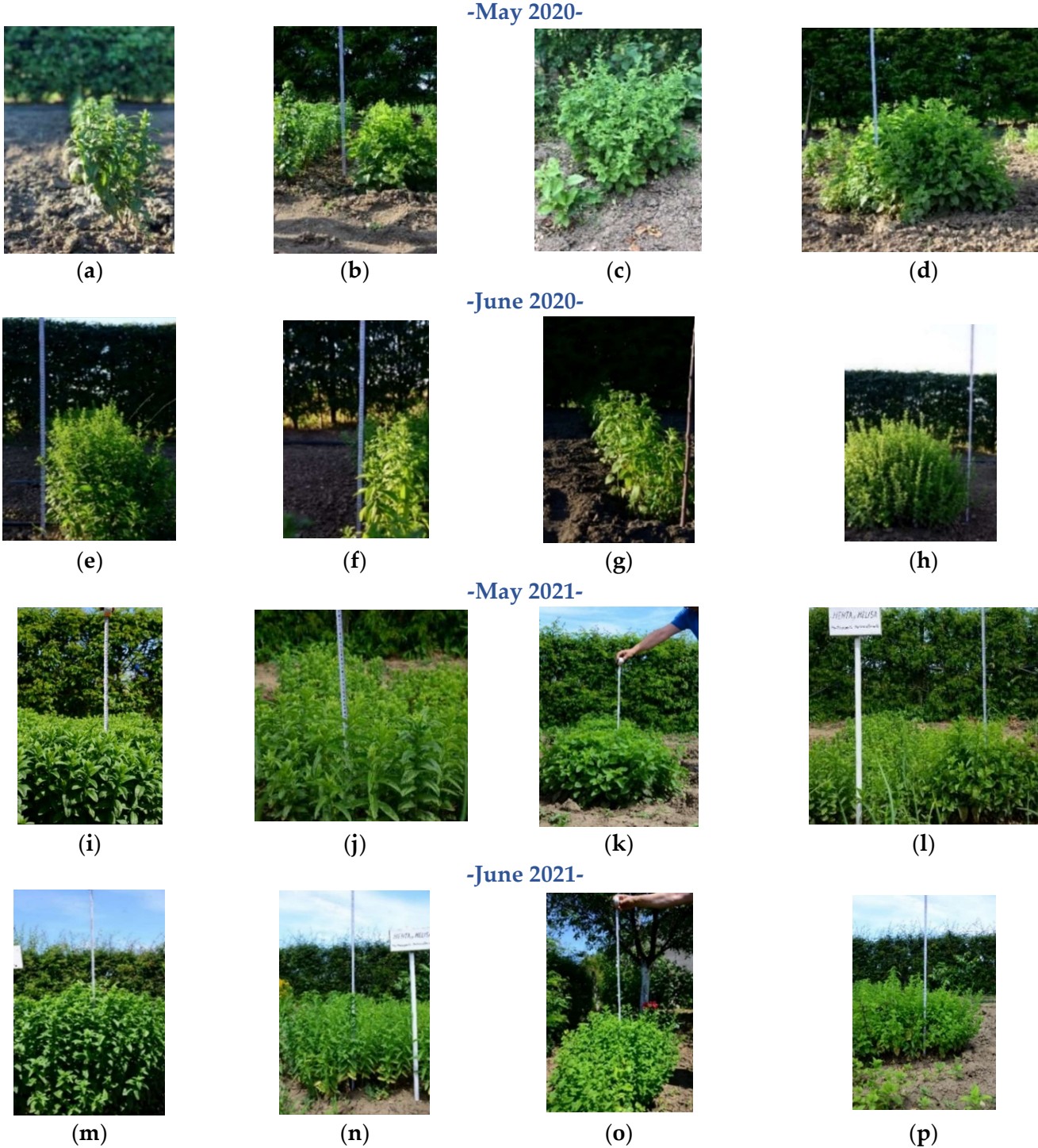

**Figure 2.** Evolutionary stages of *Mentha piperita* L. and *Melissa officinalis* L. growth in common crops compared with control ones during 2020–2021 (**a**–**p**); Legend: h—height; MM—peppermint control crop; MF—peppermint phytosociological (common) crop; MLM—lemon balm control crop; and MLF—lemon balm phytosociological (common) crop. (**a**) MM; h = 32 cm; (**b**) MF; h = 44 cm; (**c**) MLM; h = 34 cm; (**d**) MLF; h = 51 cm; (**e**) MM; h = 48 cm; (**f**) MF; h = 52 cm; (**g**) MLM; h = 55 cm; (**h**) MLF; h = 74 cm; (**i**) MM; h = 57 cm; (**j**) ML; h = 43 cm; (**k**) MLM; h = 50 cm; (**l**) MLF; h= 47 cm; (**m**) MM; h = 80 cm; (**n**) ML; h = 57 cm; (**o**) MLM; h = 70 cm; (**p**) MLF; h = 59 cm.

**Table 2.** Climatic conditions in Turnu Magurele town (Romania) during 2020–2021 according to ANM (National Weather Service) [27].

| Turnu Magurele | Period | Medium Value | Minim Value (Date) | Maxim Value (Date) | Number of Observations |
|---|---|---|---|---|---|
| T air (°C) at altitudes of 2 m | 01.05–30.06.2020 | +19.2 | +6.5 (09.05.2020) | +33.5 (29.06.2020) | 1464 |
| above the ground | 01.05–30.06.2021 | +19.5 | +4.5 (09.05.2021) | +36.8 (25.06.2021) | 1464 |
| P0, atmospheric pressure at | 01.05–30.06.2020 | 756.7 | 749.2 (02.05.2020) | 765.6 (23.05.2020) | 1464 |
| the station level (mmHg) | 01.05–30.06.2021 | 757.8 | 750.6 (13.05.2021) | 767.5 (09.05.2021) | 1464 |
| U, relative humidity (%), 2 m | 01.05–30.06.2020 | 68 | 22 (28.06.2020) | | 1464 |
| above the ground | 01.05–30.06.2021 | 69 | 23 (12.05.2021) | | 1464 |
| | | The amount of precipitation | Maxim Value (date) | The proportion of days with precipitation | Number of observations |
| RRR, the amount of precipitation (milimeters) | 01.05–30.06.2020 | 163 | 35.0 in 12 h (16.06.2020) | 23 | 122 |
| | 01.05–30.06.2021 | 135 | 22.0 in 12 h (25.06.2019) | 25 | 122 |

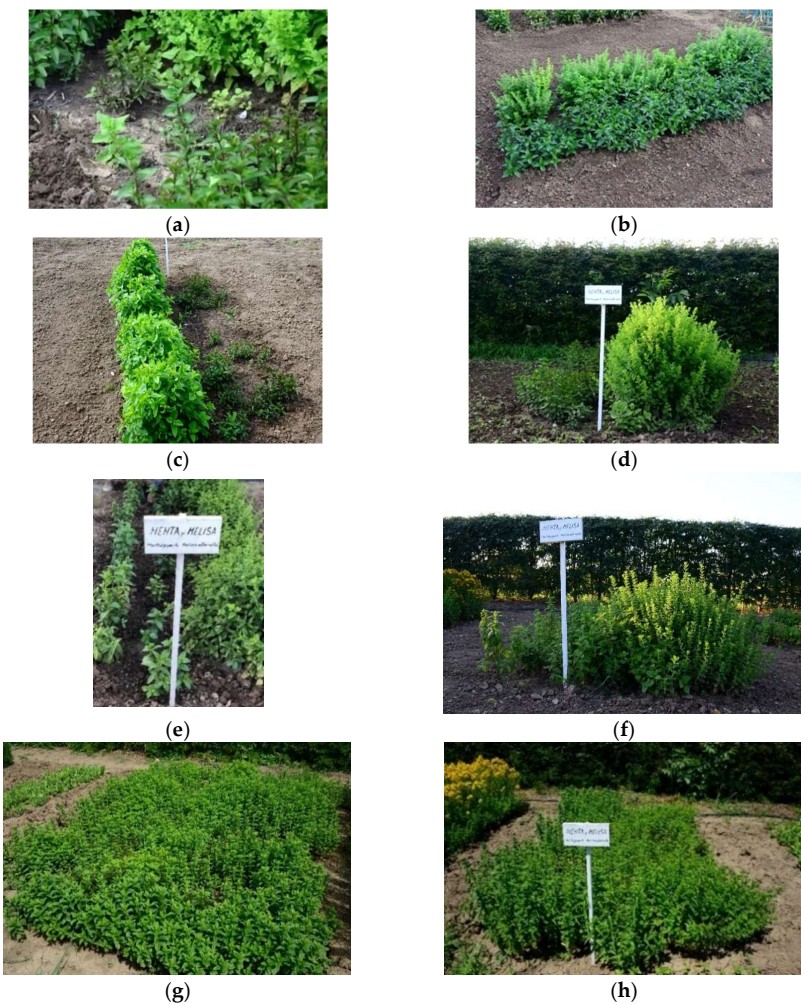

(a)    (b)

(c)    (d)

(e)    (f)

(g)    (h)

**Figure 3.** The horizontal evolution of peppermint and lemon balm phytosociological crops during 2018–2021 (**a**–**h**). (**a**) Phytosociological crop, May 2018; (**b**) Phytosociological crop, June 2018; (**c**) Phytosociological crop, May 2019; (**d**) Phytosociological crop, June 2019; (**e**) Phytosociological crop, May 2020; (**f**) Phytosociological crop, June 2020; (**g**) Phytosociological crop, May 2021; (**h**) Phytosociological crop, June 2021.

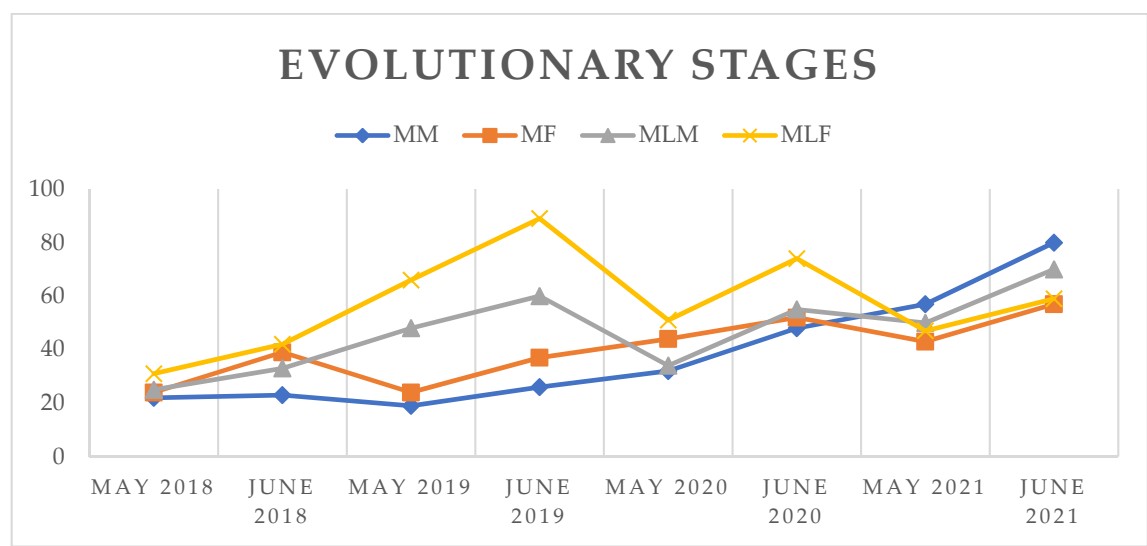

**Figure 4.** Evolutionary stages of *Mentha × piperita* L. and *Melissa officinalis* L. growth in common crops [23] compared with control ones during 2018–2021; Legend: MM—peppermint control crop; MF—peppermint phytosociological (common) crop; MLM—lemon balm control crop; and MLF—lemon balm phytosociological (common) crop; evolution of heights—y axis.

The research was extended in parallel to other species of medicinal plants, being cultivated together as follows: *Thymus vulgaris* L.–*Calendula officinalis* L. (Figure 5), *Rosmarinus officinalis* L.–*Matricaria chamomilla* L. (Figure 6a–c), *Hypericum perforatum* L.–*Chelidonium majus* L. (Figure 7a–c). All phytosociological lots (Figure 8 were compared with control lots, being grown under the same conditions.

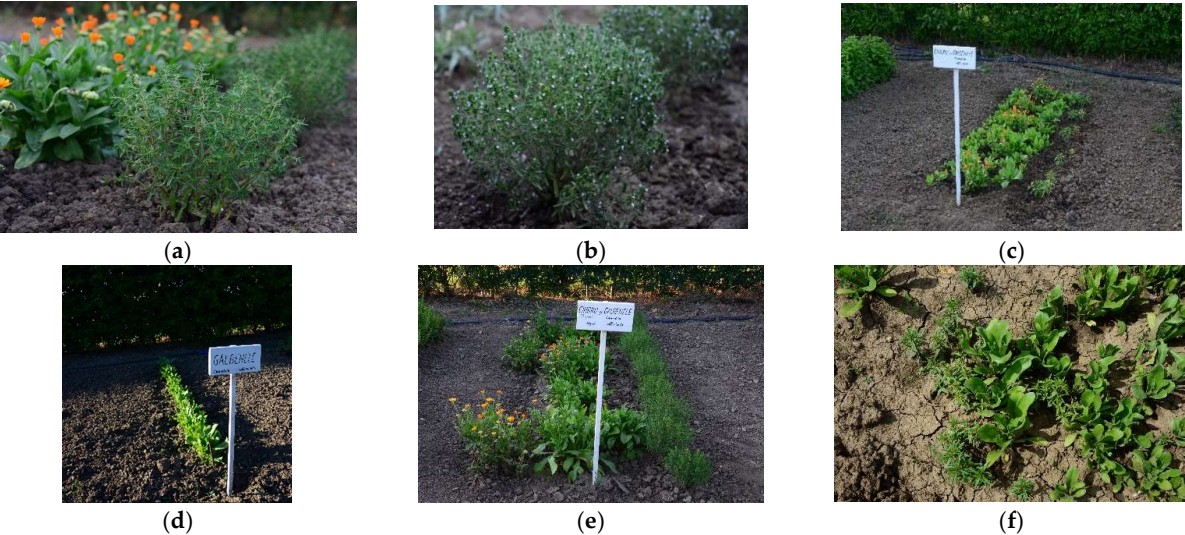

**Figure 5.** The horizontal evolution of *Thymus vulgaris* L. and *Calendula officinalis* L. phytosociological crops during 2018–2021 (**a**–**f**), where h—height. (**a**) Phytosociological crop thyme (h-23 cm) and marigolds (h-29 cm), May 2018; (**b**) Phytosocio-logical crop thyme (h-29 cm) and marigolds (h-37 cm), June 2018; (**c**) Phytosociological crop thyme (h-16 cm) and marigolds (h-33 cm), May 2019; (**d**) Control group marigolds (h-22 cm) May 2020; (**e**) Phytosociological crop thyme (h-30 cm) and marigolds (h-41 cm), June 2020; (**f**) Phytosociolog-ical crop thyme (h-14 cm) and marigolds (h-16 cm), May 2021.

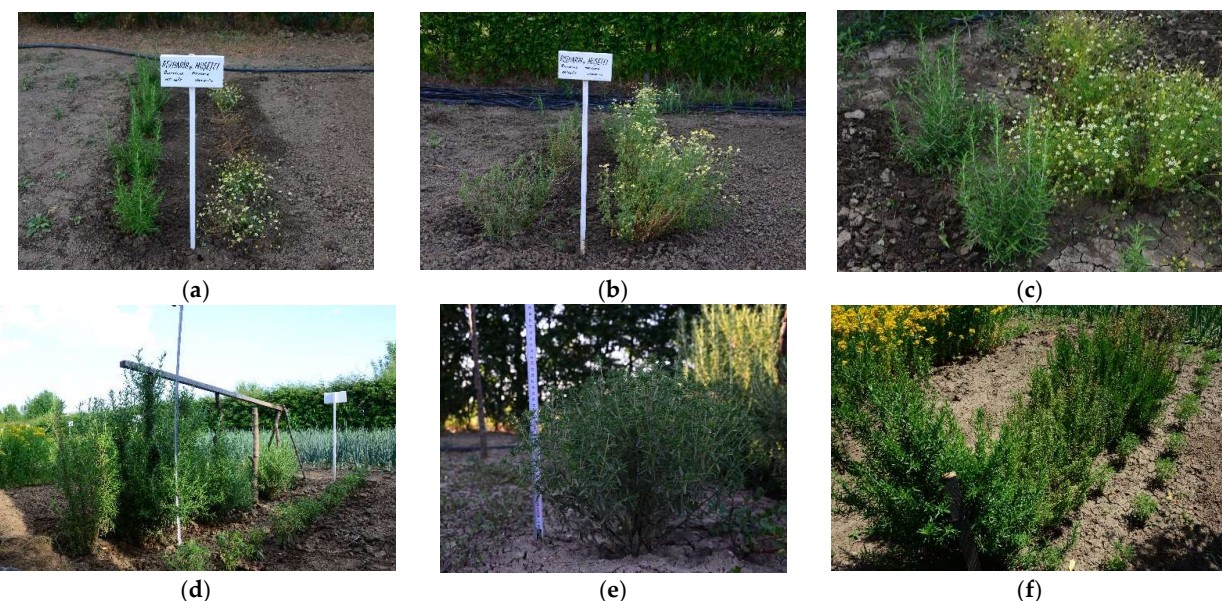

**Figure 6.** The horizontal evolution of *Rosmarinus officinalis* L. and *Matricaria chamomilla* L. phytosociological crops during 2018–2019, changed with *Rosmarinus officinalis* L. and *Thymus vulgaris* L. during 2020–2021 (**a**–**f**), where h—height. (**a**) Phytosociological crop rosemary (h-44 cm) and chamomile (h-24 cm), May 2018; (**b**) Phytoso-ciological crop rosemary (h-49 cm) and chamomile (h-54 cm), May 2019; (**c**) Phytosociological crop rosemary (h-53 cm) and chamomile (h-62 cm), June 2019; (**d**) Phytosociological crop rosemary (h-98 cm) and thyme (h-18 cm), May 2020; (**e**) Phytosociological crop rosemary (h-120 cm) and thyme (h-30 cm), June 2020; (**f**) Phytosociological crop rosemary (h-130 cm) and thyme (h-24 cm), June 2021.

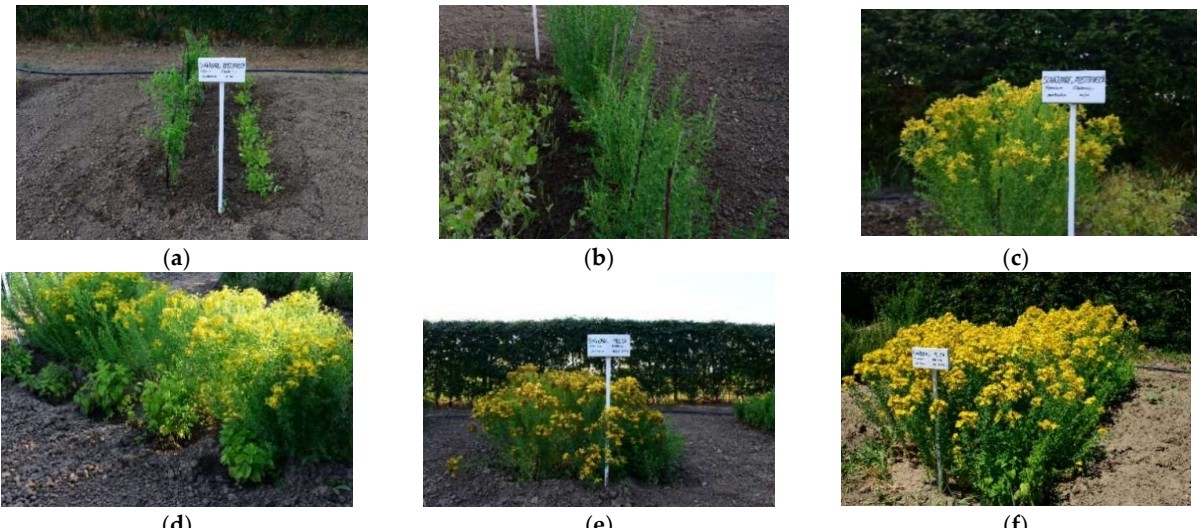

**Figure 7.** The horizontal evolution of *Hypericum perforatum* L. and *Chelidonium majus* L. phytosociological crops during 2018–2019 (**a**–**f**), changed with *Hypericum perforatum* L. and *Melissa officinalis* L. during 2020–2021, where h—height. (**a**) Phytosociological crop St. John's Wort (h-60 cm) and celandine (h-25 cm), May 2018; (**b**) Phy-tosociological crop St. John's Wort (h-73 cm) and celandine (h-52 cm), May 2019; (**c**) Phytosocio-logical crop St. John's Wort (h-93 cm) and celandine (h-58 cm) June 2019; (**d**) Phytosociological crop St. John's Wort (h-74 cm) and lemon balm (h-28 cm), May 2020; (**e**) Phytosociological crop St. John's Wort (h-78 cm) and lemon balm (h-52 cm), June 2020; (**f**) Phytosociological crop St. John's Wort (h-82 cm) and lemon balm (h-56 cm), June 2021.

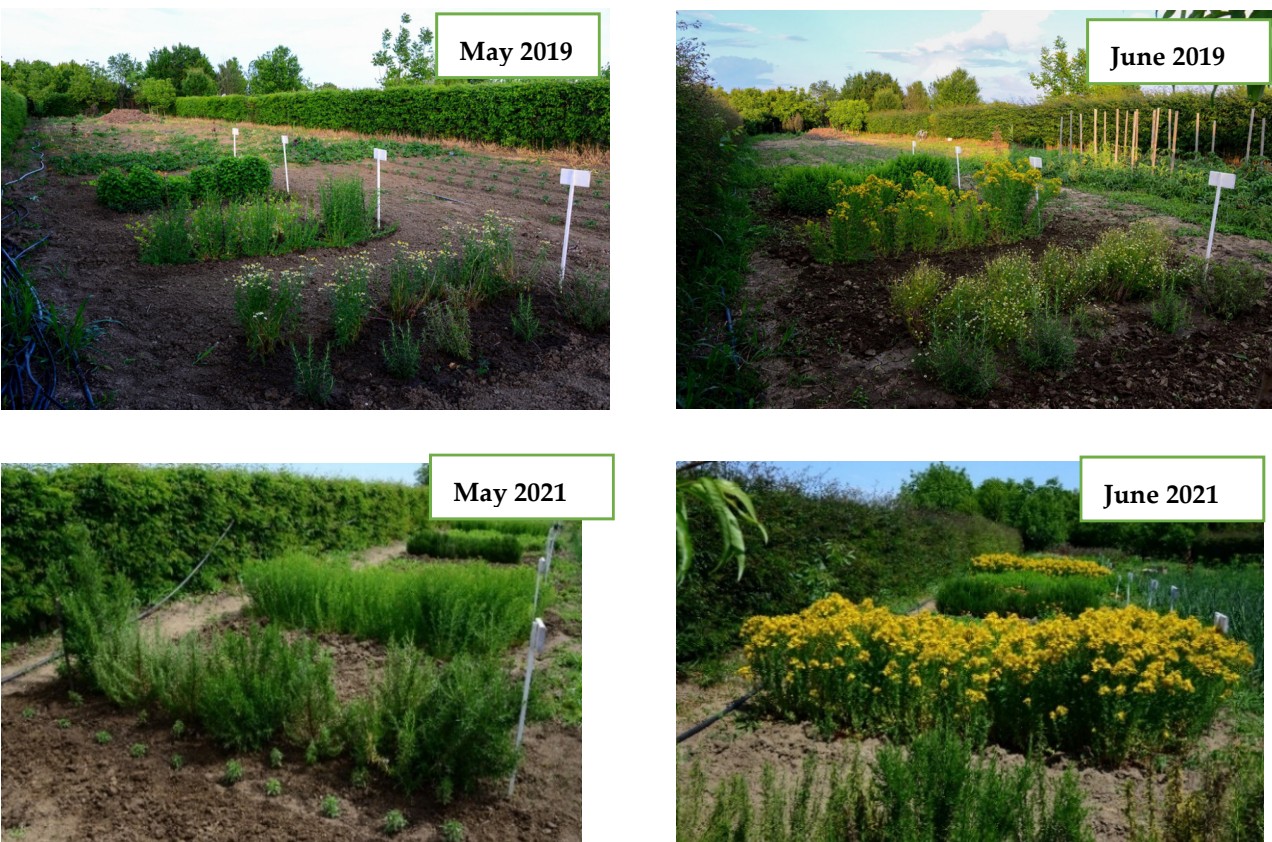

**Figure 8.** The evolution and dynamics of the whole culture during our study.

Since the association between rosemary and chamomile did not lead to the expected results, we decided to introduce into the culture the association between rosemary and thyme during 2020–2021 (Figure 6d–f).

Considering that the St. John's Wort has a different period of development compared to the hollyhock (Figure 7a–c), we decided to set up a new batch for further research. Therefore, in 2020, we cultivated St. John's Wort with another medicinal plant in a common crop–*Melissa officinalis* L. (Figure 7d–f).

Figure 8 represents a panoramic characterization of the evolution of the crops in 2019–2021, and Figure 9 shows graphically the evolution of the rest of medicinal plants for the entire study period.

In order to observe the morphological changes and certain anatomical features, we also used a macro and microscopic examination using a stereomicroscope (ZEISS Stemi 508 Greenough Stereo Microscope with 8:1 Zoom, details up to 50× magnification) in which we analyzed different types of plant organs (flowers, stems, and leaves).

After analyzing the freshly harvested plant material, differences were found between the plant product from the phytosociological crop and from the control one. In the samples of mint and lemon balm (flowers and leaves) there were numerous octocellular glandular hairs that accumulate volatile oil compared to the control crop, where they were less widespread; for the lemon balm in the phytosociological crop, there were crowded and more developed tector hairs (Figure 10).

In the thyme leaves and flowers grown in a common crop with marigolds, a higher density of glandular bristles can be observed, which would imply a higher secretion of volatile oil. In the marigold flowers from the common sample, an abundance of cells containing stratified pigment are found, possibly tetraterpenic derivatives of a carotenoid type (Figure 11) [28].

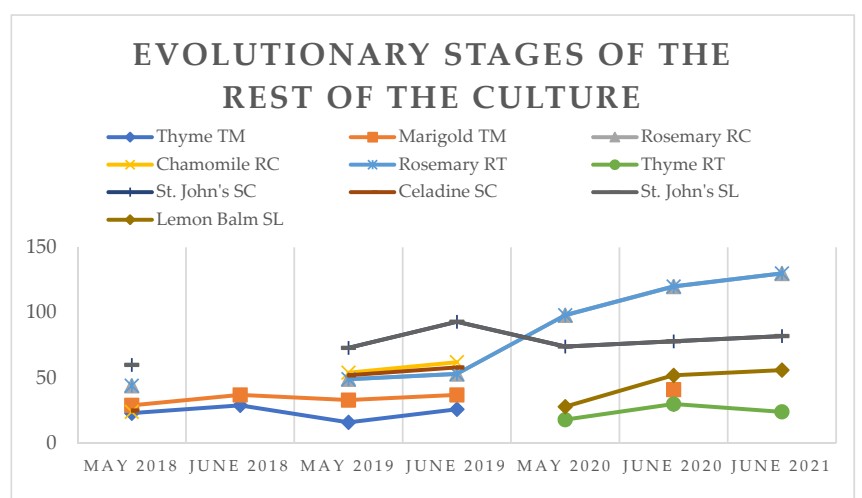

**Figure 9.** Evolutionary stages of the phytosociological culture; Legend: Thyme and Marigold (TM)—*Thymus vulgaris* L.–*Calendula officinalis* L. crop, Rosemary and Chamomile (RC)—*Rosmarinus officinalis* L.–*Matricaria chamomilla* L. crop, Rosemary and Thyme (RT)—*Rosmarinus officinalis* L.–*Thymus vulgaris* L. crop, St. John's and Celadine (SC)—*Hypericum perforatum* L.–*Chelidonium majus* L. crop, St. John's and Lemon Balm (SL)—*Hypericum perforatum* L.–*Melissa officinalis* L. crop; evolution of heights—y axis.

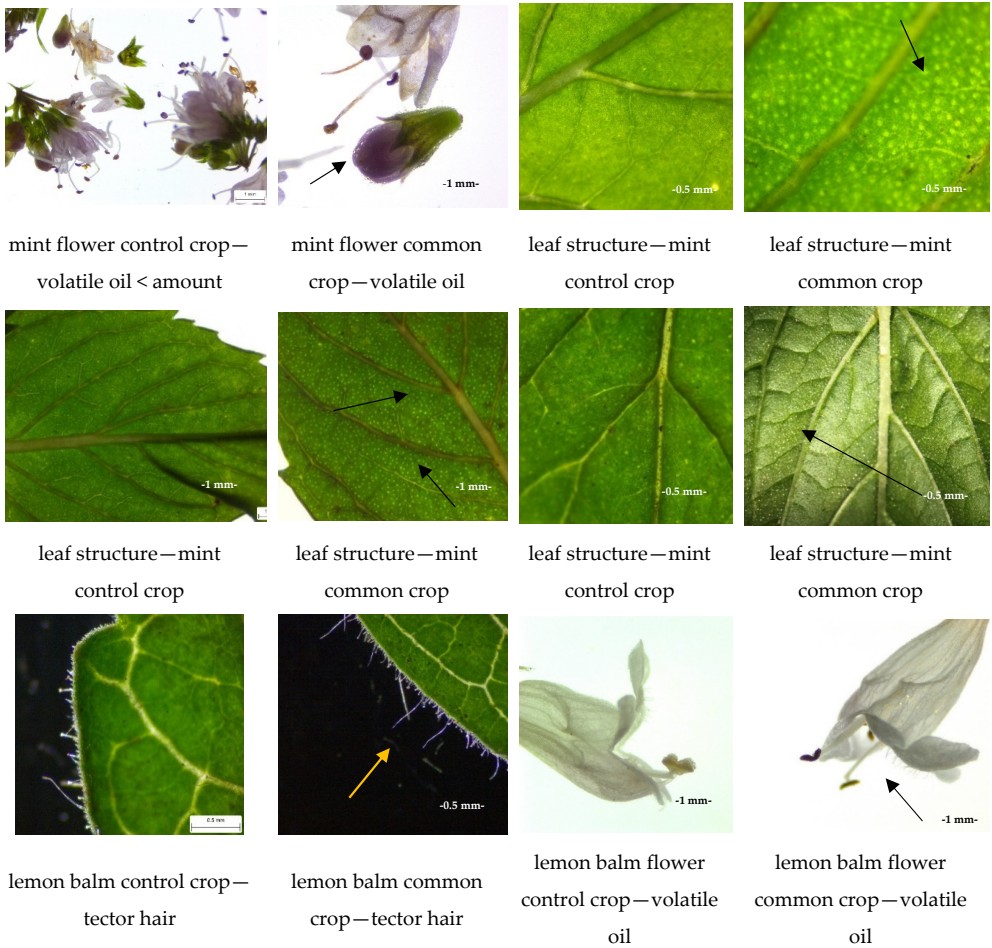

**Figure 10.** Mint—Lemon balm.

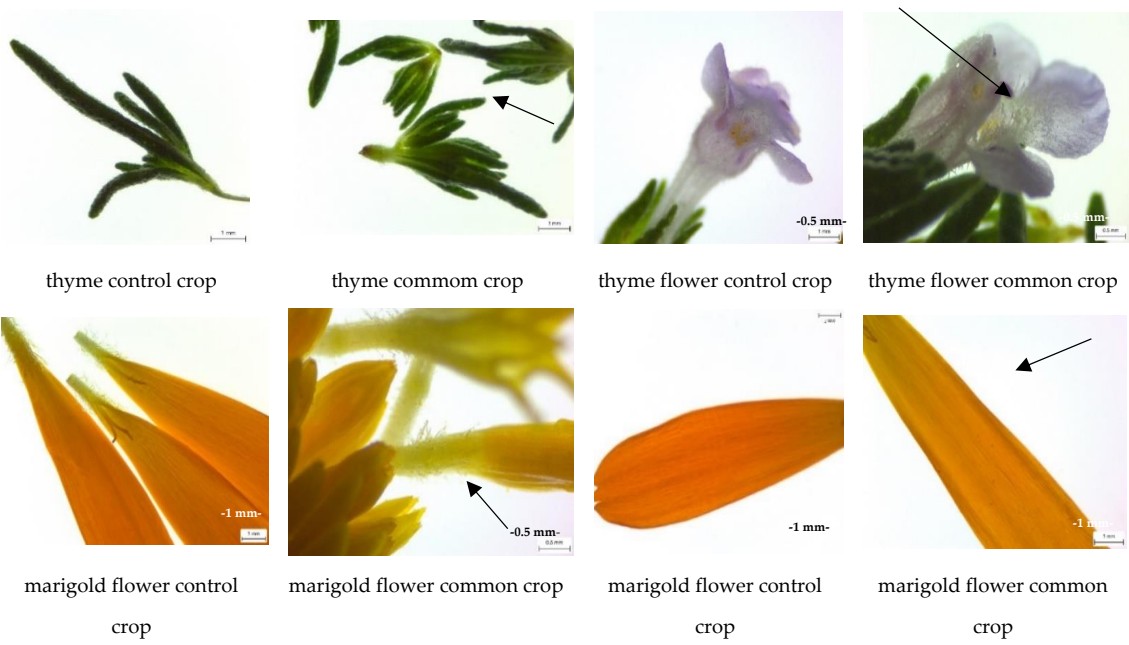

**Figure 11.** Thyme—Marigold.

The rosemary-thyme group had the following changes - the rosemary leaf had a higher number of glandular hairs with volatile oil than the control crop, and the thyme leaf and flower exhibited similar changes (Figure 12).

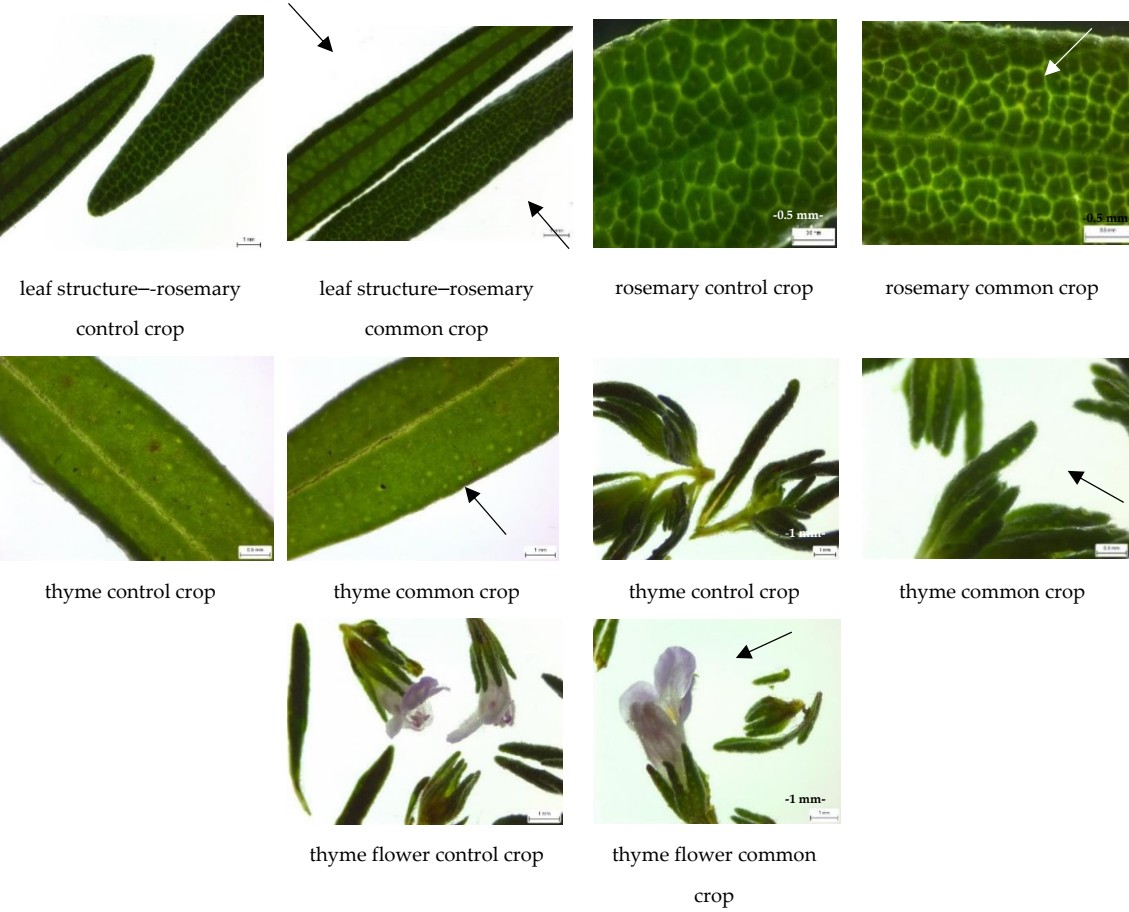

**Figure 12.** Rosemary—Thyme.

The leaves and petals of St. John's Wort flowers in the studied crop show a higher number of secretory bags with hypericin (expressed by an increased number of black dots located in the upper peripheral area), and lemon balm leaves have a visibly higher number of octocellular glandular hairs (Figure 13) compared to the control crop.

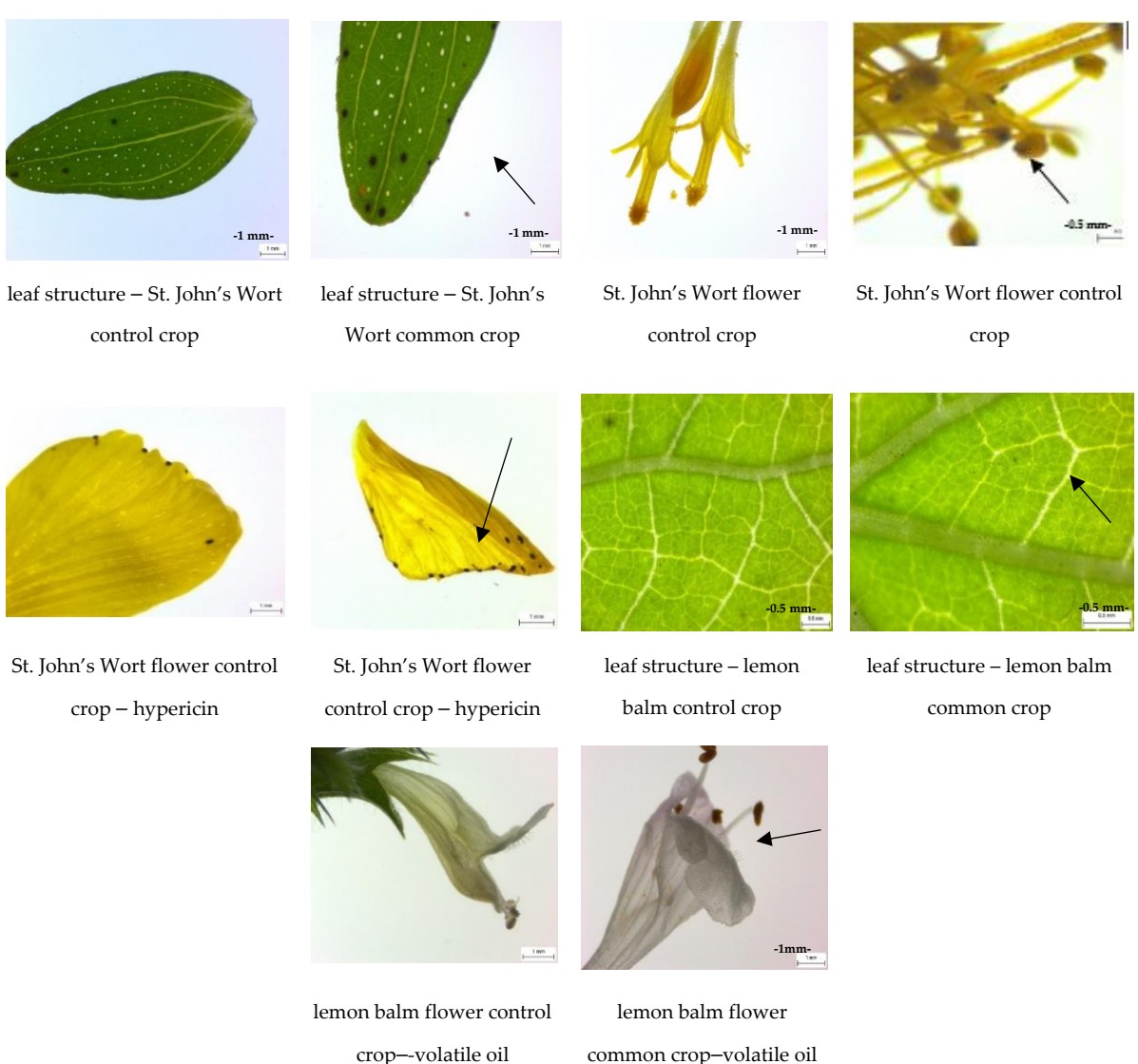

| | |
|---|---|
| leaf structure – St. John's Wort control crop | leaf structure – St. John's Wort common crop |
| St. John's Wort flower control crop | St. John's Wort flower control crop |
| St. John's Wort flower control crop – hypericin | St. John's Wort flower control crop – hypericin |
| leaf structure – lemon balm control crop | leaf structure – lemon balm common crop |
| lemon balm flower control crop–-volatile oil | lemon balm flower common crop–volatile oil |

**Figure 13.** St. John's Wort—Lemon Balm.

Another important difference appeared in the yarrow-chamomile phytosociological group, with a more obvious horizontal development of the yarrow leaves compared to the control group (Figure 14).

From the point of view of the quantity of vegetable product supplied by each lot, there is a significant difference in all the samples analyzed compared to the control ones. (Figures 15 and 16—Table 4). We want to mention that for mint, lemon balm and rosemary—the leaves were weighed; for thyme, St. John's Wort, chamomile and yarrow—the upper third of the aerial part and for marigolds—the flowers were weighed.

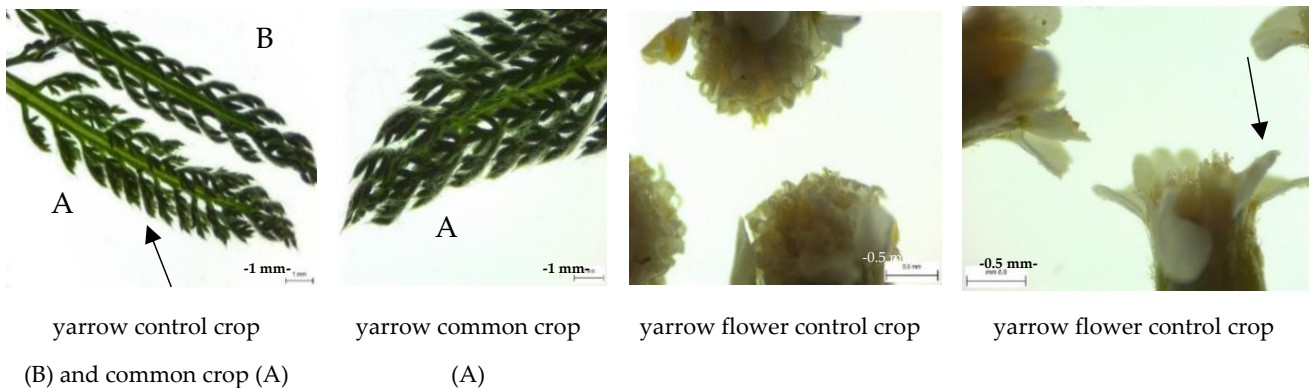

yarrow control crop
(B) and common crop (A)

yarrow common crop
(A)

yarrow flower control crop

yarrow flower control crop

**Figure 14.** Yarrow.

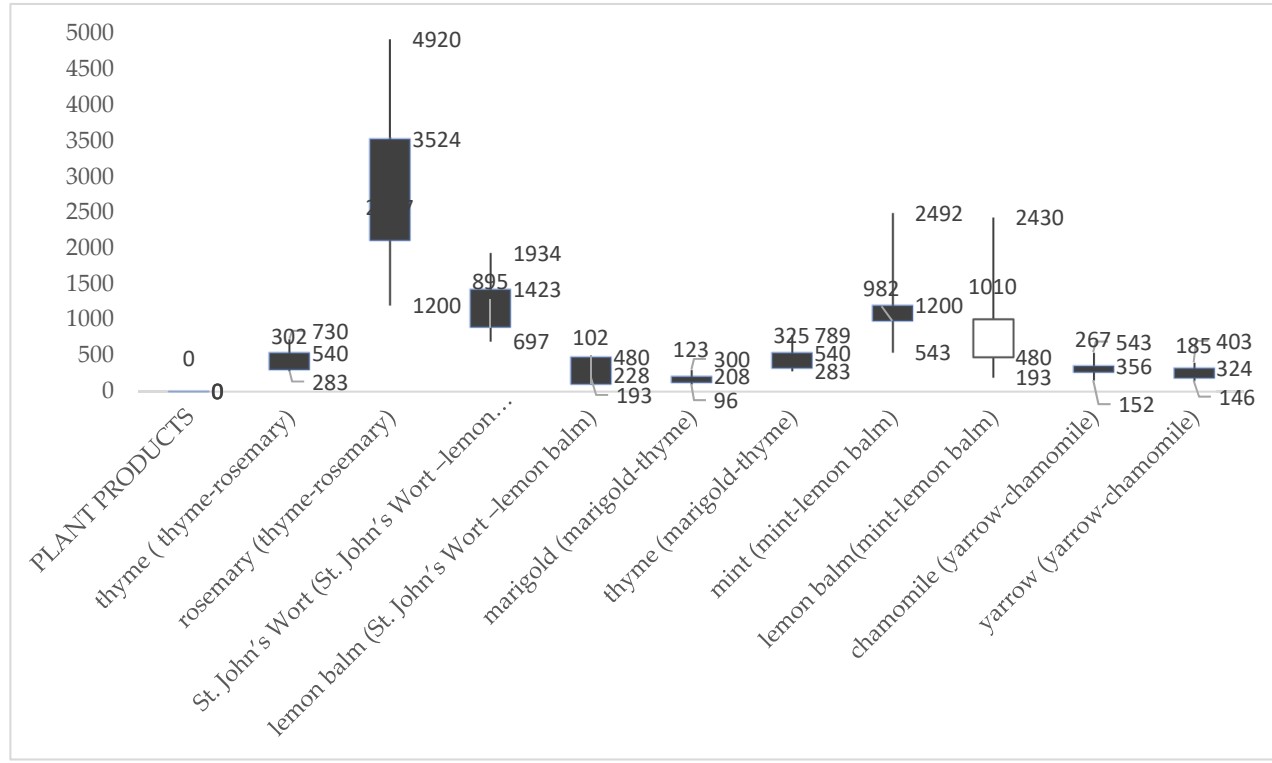

**Figure 15.** Graphic representation of vegetable products' mass in 2020.

**Table 3.** Representation of vegetable products' mass in 2020.

| Plant Products | Control Crop | | Common Crop | |
|---|---|---|---|---|
| | g Harvested | g Dry | g Harvested | g Dry |
| thyme (thyme-rosemary) | 540 | 283 | 730 | 302 |
| rosemary (thyme-rosemary) | 3524 | 1200 | 4920 | 2107 |
| St. John's Wort (St. John's Wort—lemon balm) | 1423 | 697 | 1934 | 895 |
| lemon balm (St. John's Wort—lemon balm) | 480 | 193 | 228 | 102 |
| marigold (marigold-thyme) | 208 | 96 | 300 | 123 |
| thyme (marigold-thyme) | 540 | 283 | 789 | 325 |
| mint (mint-lemon balm) | 1200 | 543 | 2492 | 982 |
| lemon balm (mint-lemon balm) | 480 | 193 | 2430 | 1010 |
| chamomile (yarrow-chamomile) | 356 | 152 | 543 | 267 |
| yarrow (yarrow-chamomile) | 324 | 146 | 403 | 185 |

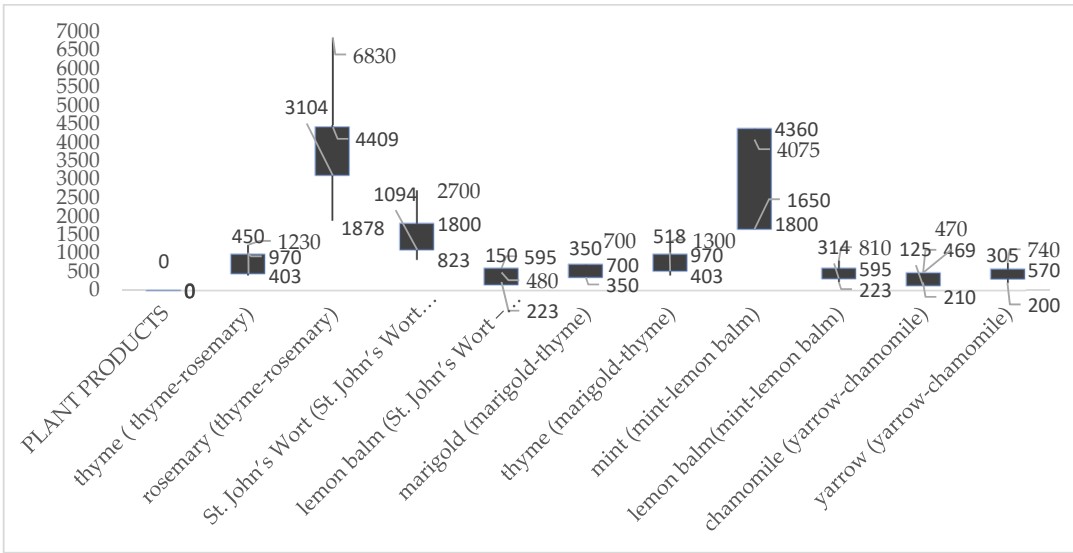

**Figure 16.** Graphic representation of vegetable products mass in 2021.

**Table 4.** Representation of vegetable products' mass in 2021.

| Plant Products | Control Crop | | Common Crop | |
|---|---|---|---|---|
| | g Harvested | g Dry | g Harvested | g Dry |
| thyme (thyme-rosemary) | 970 | 403 | 1230 | 450 |
| rosemary (thyme-rosemary) | 4409 | 1878 | 6830 | 3104 |
| St. John's Wort (St. John's Wort—lemon balm) | 1800 | 823 | 2700 | 1094 |
| lemon balm (St. John's Wort—lemon balm) | 595 | 223 | 480 | 150 |
| marigold (marigold-thyme) | 700 | 350 | 700 | 350 |
| thyme (marigold-thyme) | 970 | 403 | 1300 | 518 |
| mint (mint-lemon balm) | 4360 | 1800 | 4075 | 1650 |
| lemon balm(mint-lemon balm) | 595 | 223 | 810 | 314 |
| chamomile (yarrow-chamomile) | 469 | 210 | 470 | 125 |
| yarrow (yarrow-chamomile) | 570 | 200 | 740 | 305 |

According to the data analysis obtained from Tables 3 and 4 and following the graphical representation in Figures 16 and 17, there has been a development in providing the amount of vegetable raw material. It is much more obvious in phytosociological crops other than the control ones.

The results of the quantitative chemical determinations are given in Table 5 (volumetric dosing of volatile oil) and Table 6 (dosing of flavones, PCAs and total phenolic content through spectrophotometric methods). These data are represented graphically in Figure 17 (representation of total flavones content) and Figures 18 and 19 constitute the total PCAs content respectively TPC.

**Table 5.** Determining the amount of essential oil in certain vegetable products.

| | mL Essential Oil/100 g Dry Herbal Product | |
|---|---|---|
| | Control Crop | Common Crop |
| rosemary (thyme-rosemary) | 3.6 | 4 |
| thyme (thyme-rosemary) | 3.6 | 6.6 |
| thyme (marigold-thyme) | 3.6 | 5 |
| mint (mint-lemon balm) | 1.16 | 1.25 |
| lemon balm (mint-lemon balm) | 0.6 | 2 |
| yarrow (yarrow-chamomile) | 0.4 | 0.6 |
| chamomile (yarrow-chamomile) | 0.2 | 0.3 |

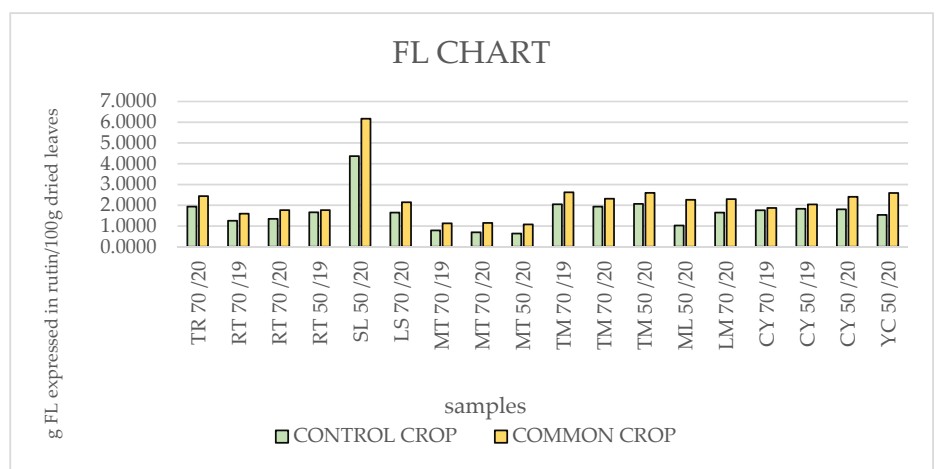

**Figure 17.** Graphic representation of total flavones content.

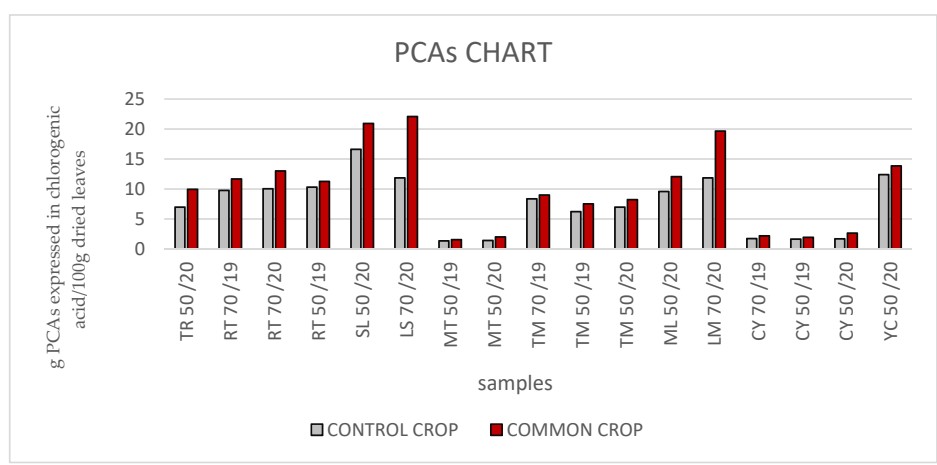

**Figure 18.** Graphic representation of total PACs (phenolcarboxylic acids) content.

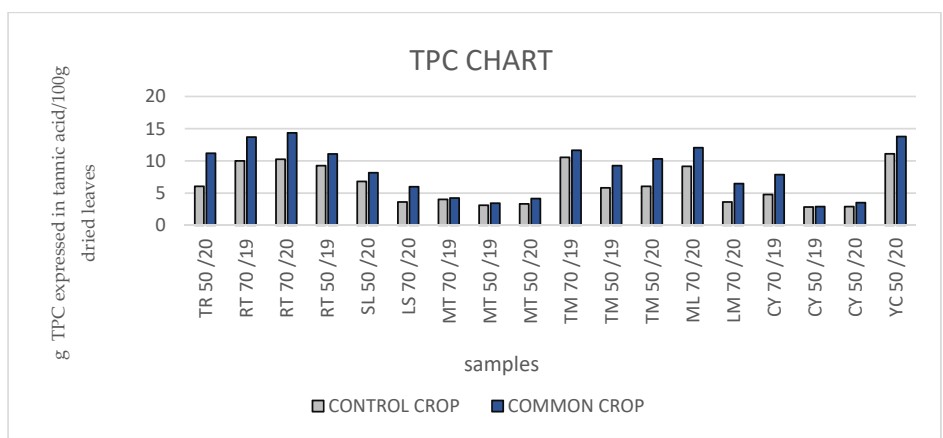

**Figure 19.** Graphic representation of total phenolic content. As we highlight in the figures for the each Plant product (Figure 20**a**–**j**) there is a statistically significant interaction ($p$ value < 0.05) between the effects of Compound and Sample on value of concentration except four cases: MT, TM, ML and YC. Simple main effects analysis showed that the common crop is statistically different from control crop ($p$ value < 0.05), except YC and between FL, PCA's and TPC there are statistical differences, except TR, TM, ML and YC where PCA's and TPC behave similarly statistically.

**Table 6.** Results for spectrophotometric and volumetric assay for every medicinal plant in the culture.

| Plant Product | Solvent | g FL Expressed in Rutin/100 g Dried Leaves | | | | g PCAs Expressed in Chlorogenic Acid/100 g Dried Leaves | | | | g TPC Expressed in Tannic Acid/100 g Dried Leaves | | | |
|---|---|---|---|---|---|---|---|---|---|---|---|---|---|
| | | Control Crop | | Common Crop | | Control Crop | | Common Crop | | Control Crop | | Common Crop | |
| | Alcohol | 2019 | 2020 | 2019 | 2020 | 2019 | 2020 | 2019 | 2020 | 2019 | 2020 | 2019 | 2020 |
| TR | 70% | - | 1.9317 ± 0.0947 | - | 2.4413 ± 0.1858 | - | - | - | - | - | - | - | - |
| | 50% | - | - | - | - | - | 6.9709 ± 1.4921 | - | 9.9461 ± 0.8385 | - | 6.0393 ± 0.3204 | - | 11.1911 ± 0.7959 |
| RT | 70% | 1.2555 ± 0.3082 | 1.3469 ± 0.1941 | 1.5908 ± 0.1292 | 1.7616 ± 0.1322 | 9.7633 ± 0.3391 | 10.0288 ± 0.4307 | 11.659 ± 1.1725 | 13.0085 ± 0.5305 | 10.0337 ± 0.2470 | 10.2605 ± 0.4612 | 13.6982 ± 3.4303 | 14.3533 ± 3.4511 |
| | 50% | 1.6612 ± 0.2336 | - | 1.7626±0.2195 | - | 10.312 ± 1.4714 | - | 11.2637 ± 1.4027 | - | 9.2616 ± 0.3351 | - | 11.0854 ± 0.2787 | - |
| SL | 70% | - | - | - | - | - | - | - | - | - | - | - | - |
| | 50% | - | 4.3646 ± 1.4447 | - | 6.1703 ± 1.1658 | - | 16.6146 ± 1.0430 | - | 20.9229 ± 0.9239 | - | 6.7989 ± 0.3940 | - | 8.1598 ± 0.4262 |
| LS | 70% | - | 1.6432 ± 0.2505 | - | 2.1422 ± 0.5379 | - | 11.8405 ± 0.7671 | - | 22.0896 ± 1.5231 | - | 3.614 ± 0.421 | - | 5.9761 ± 0.0938 |
| | 50% | - | - | - | - | - | - | - | - | - | - | - | - |
| MT | 70% | 0.787 ± 0.1351 | 0.6944 ± 0.0805 | 1.1311 ± 0.0578 | 1.1504 ± 0.0643 | - | - | - | - | 4.0237 ± 0.8222 | - | 4.2247 ± 1.6928 | - |
| | 50% | 0.6376 ± 0.0505 | - | 1.0759 ± 0.0951 | - | 1.3438 ± 0.0999 | 1.4104 ± 0.1216 | 1.5514 ± 0.1935 | 2.0048 ± 0.2633 | 3.1223 ± 0.2800 | 3.3329 ± 0.4030 | 3.4311 ± 0.7578 | 4.1516 ± 0.5974 |
| TM | 70% | 2.0462 ± 0.5865 | 1.9317 ± 0.0947 | 2.6249 ± 1.1390 | 2.3134 ± 0.4572 | 8.3479 ± 1.3352 | - | 8.9926 ± 1.0686 | - | 10.556 ± 1.3394 | - | 11.639 ± 2.2604 | - |
| | 50% | 2.0646 ± 0.2753 | - | 2.5947 ± 0.0961 | - | 6.2302 ± 0.9905 | 6.9709 ± 1.4921 | 7.5046 ± 0.2743 | 8.2233 ± 0.5946 | 5.8147 ± 1.0630 | 6,0393 ± 0.3204 | 9.2512 ± 1.3221 | 10.3147 ± 1.2546 |
| ML | 70% | - | - | - | - | - | - | - | - | - | 9.1505 ± 1.9447 | - | - |
| | 50% | - | 1.024 ± 0.2407 | - | 2.2621 ± 0.1475 | - | 9.5829 ± 1.0670 | - | 12.0579 ± 0.7928 | - | - | - | 12.041 ± 1.9260 |
| LM | 70% | - | 1.6432 ± 0.2505 | - | 2.2951 ± 0.7055 | - | 11.8405 ± 0.7671 | - | 19.6639 ± 2.6681 | - | 3.614 ± 0.421 | - | - |
| | 50% | - | - | - | - | - | - | - | - | - | - | - | 6.4694 ± 0.5147 |
| CY | 70% | 1.7606 ± 0.1229 | - | 1.8712 ± 0.1004 | - | 1.7185 ± 0.2359 | - | 2.1868 ± 0.2834 | - | 4.7869 ± 0.6933 | - | 4.8703 ± 1.1159 | - |
| | 50% | 1.8272 ± 0.5233 | 1.8057 ± 0.5497 | 2.0404 ± 0.6936 | 2.4074 ± 1.3468 | 1.629 ± 0.2360 | 1.682 ± 0.2470 | 1.9253 ± 0.1624 | 2.6278 ± 0.3760 | 2.8405 ± 0.2988 | 2.8963 ± 0.3025 | 2.9047 ± 0.2621 | 3.5099 ± 0.2954 |
| YC | 70% | - | - | - | - | - | - | - | - | - | - | - | - |
| | 50% | - | 1.5354 ± 0.1772 | - | 2.5909 ± 0.4796 | - | 12.4033 ± 4.5895 | - | 13.8511 ± 4.3555 | - | 11.1061 ± 0.8620 | - | 13.7817 ± 2.9323 |

Results are mean ± SD ($n = 3$); FL—flavones; PCAs—phenolcarboxylic acids; TPC—total phenolic content; Legend: TR—thyme (thyme-rosemary); RT—rosemary (thyme-rosemary); SL—St. John's Wort (St. John's Wort—lemon balm); LS—lemon balm (St. John's Wort—lemon balm); MT—marigold (marigold-thyme); TM—thyme (marigold-thyme); ML—mint (mint-lemon balm); LM—lemon balm(mint-lemon balm); CY—chamomile (yarrow-chamomile); YC—yarrow (yarrow-chamomile).

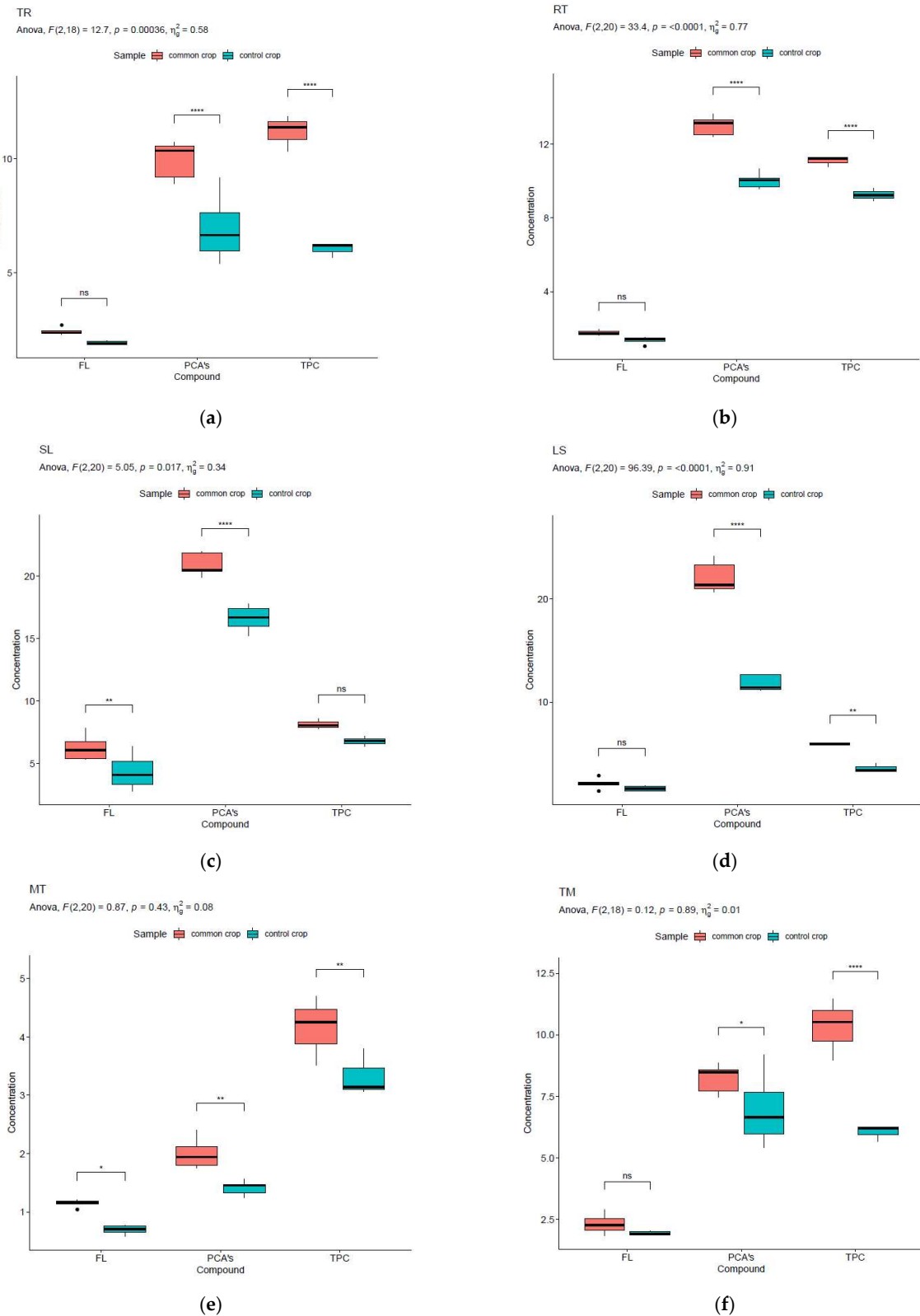

**Figure 20.** *Cont.*

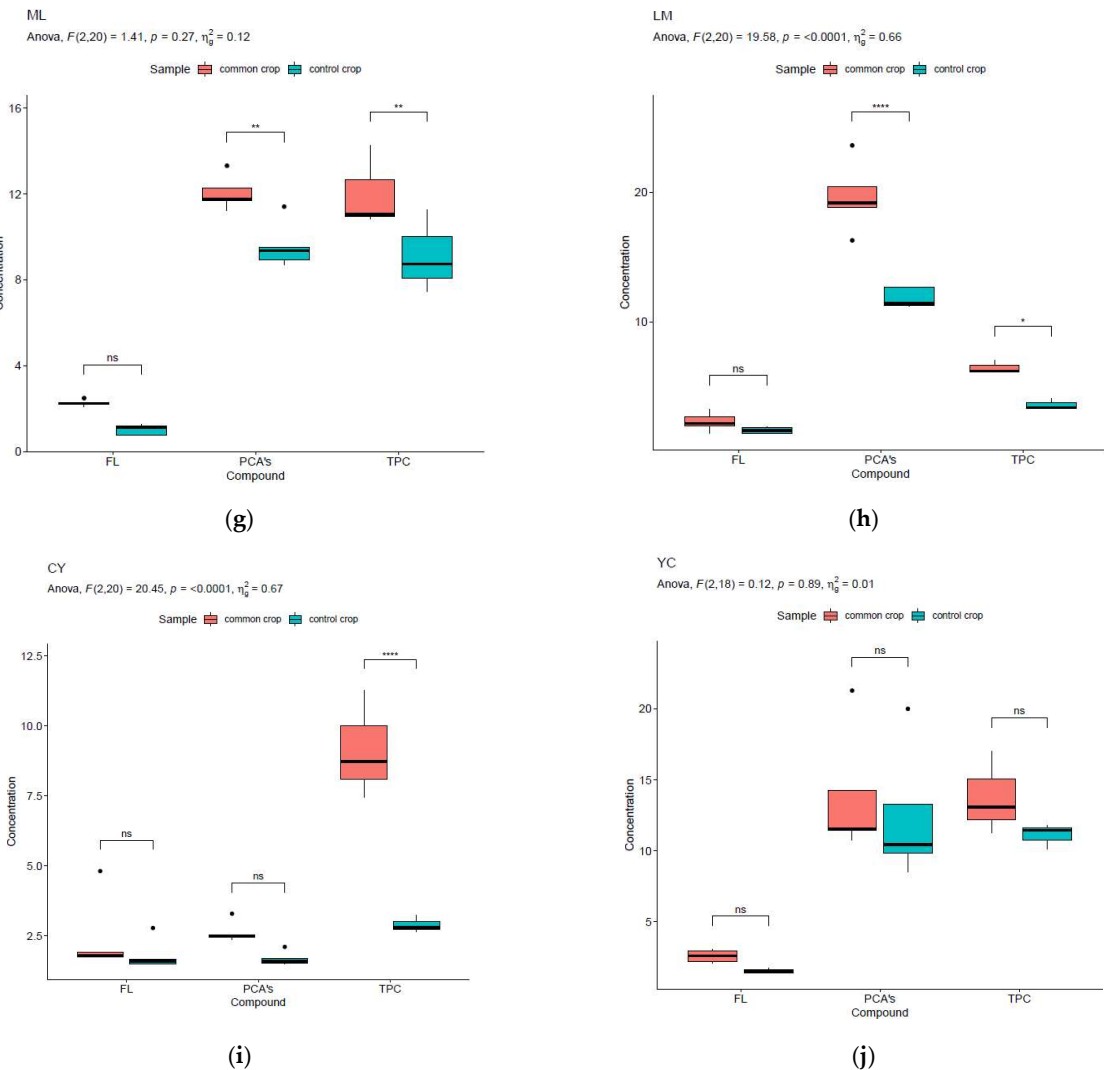

**Figure 20.** Statistical analysis a two-way interaction boxplot for medicinal plant. (**a**) A two-way interaction boxplot for TR; (**b**) A two-way interaction boxplot for RT; (**c**) A two-way interaction boxplot for SL; (**d**) A two-way interaction boxplot for LS; (**e**) A two-way in-teraction boxplot for MT; (**f**) A two-way interaction boxplot for TM; (**g**) A two-way interaction boxplot for ML; (**h**) A two-way interaction boxplot for LM; (**i**) A two-way interaction boxplot for CY; (**j**) A two-way interaction boxplot for YC.

## 4. Discussion

In this present research we have been observing this relationship and how a group of medicinal plants can be influenced, by being grown together. This activity took place without intervenion in any way for the common crop such as: enriching the soil with a type of fertilizer, applying different amounts of water than the control one, etc. Morpho-anatomical differences occurred in the distribution of frequency, density, abundance and relative frequency in the common group compared to the control group. Compared to the control crops, it was observed that each combination of medicinal plants leads to the biosynthesis of a larger quantity of secondary metabolites (volatile oil, flavones, PCAs and total polyphenols—Table 5 and Figures 17–19). This was also remarked ino the study published on the association between mint and lemon balm, the culture period followed being 2018–2019 [23]. Based on the statistical analysis, a direct correlation was found in certain types of cultures in terms of the biosynthesis of secondary metabolites. Following the same batches of mint and lemon balm during 2020–2021, showed a net increase in the production of plant raw materials, a fact possibly due to weather conditions, but also much

better accommodation in the culture of the two medicinal species. Although they reacted well in different associations, there were also situations of medicinal plants that did not adapt (*Rosmarinus officinalis* L. and *Matricaria chamomilla* L.), perhaps due to the different period of development (*Hypericum perforatum* L. and *Chelidonium majus* L.).

These are the observations of our study, unfortunately without correlating with the literature data, as the published data generally refer to the associations of plants that grow in spontaneous flora and forest areas [29,30].

A higher dominance of octocellular glandular bristles in plant products from Lamiaceae family species is correlated according to our research with a slightly higher amount of volatile oil, the quantities being dependent on the nature of the plant's raw material. The presence of cells with dominant pigments in the epidermal cells derived from petals in marigold flowers is closely correlated with a higher content of flavonic derivatives, yellow pigments. According to the results obtained from the quantitative chemical determinations, it is found that ethanol 50% extracts a larger amount of active ingredients, which can be a starting point in the more detailed phytochemical analyses performed on these plant species.

## 5. Conclusions

Following the study, we can say that the combinations of medicinal plants, either belonging to the same family or different families, are beneficial, as a consequence of their horizontal and vertical development compared to control crops, the amount of plant product provided being much higher, including the anatomical formations that accumulate volatile oil being dominant in these associations.

We mention the fact that at the moment we are studying other common crops of mint-lemon balm where we aim to measure the quantity and quality of microelements following the enrichment of the soil with a chemical or biological fertilizer. We will compare them with a control soil.

**Author Contributions:** Conceptualization: E.A.L., C.E.G. and M.G. Data curation: E.A.L. and C.E.G.; Formal analysis: E.A.L. and C.E.G.; Investigation: E.A.L., C.E.G. and M.G.; Methodology: E.A.L., C.E.G. and M.G.; Supervision: C.E.G. and M.G.; Validation: C.E.G.; Visualization: E.A.L. and C.E.G.; Writing—original draft preparation: E.A.L. and C.E.G.; Writing—review and editing: E.A.L., C.E.G. and M.G.; Project administration: E.A.L. and C.E.G. All authors have read and agreed to the published version of the manuscript.

**Funding:** This research received no external funding.

**Data Availability Statement:** MDPI Research Data Policies at https://www.mdpi.com/ethics (5 December 2021).

**Acknowledgments:** This research received no specific grant from any funding agency in the public, commercial, or not-for-profit sectors. All individuals included in this section have consented to the acknowledgement.

**Conflicts of Interest:** The authors declare that there are no conflict of interest related to this article.

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
