# Peer review of "The Initiation of a Phytosociological Study on Certain Types of Medicinal Plants"

_agriculture, doi:10.3390/agriculture12020283_

Round 1
Reviewer 1 Report
Dear Authors,
Although the idea and implementation of this project are good, I have some comments that need to be considered. Namely,:
- Correlating the essential oil content with the number of glandular hairs is risky. Since you haven't done the distillation with the Dering apparatus, how can you be sure that there was more essential oil in the test sample vs control? Without such results, you must not even assume that it is so.
- Higher plant biomass doesn't guarantee a higher content of secondary metabolites. Why the assumption that there will be more phytochemicals from the higher biomass?
Therefore, I think that in order to accurately assess the value of the investigated medicinal plants, at least the volumes of isolated essential oils and/or at least the total content of e.g. polyphenols should be examined. This will allow a broader and more precise conclusion (without any speculation), especially in the area of ​​allelopathy.
Best regards
Author Response
Dear reviewer, thank you for taking the time to evaluate this manuscript. Your feedback is appreciated. Please find attached the document with our answers to your comments / suggestions and the revised manuscript.
In order to accurately assess the value of the investigated medicinal plants, we incluted the volumes of isolated essential oils and at the total content of polyphenols. We excluded the discution about allelopathy.
Reviewer 2 Report
I have critically reviewed the manuscript, the manuscript methodology is very poorly explain and study is not supported by the scientific literature.
I have main concern over study the data, most of the data in the form of pictures, and just evaluation the growth of different medicinal plants is not valid, as we require their active ingredients, for which authors have not collected the data.
Author Response
Dear reviewer, thank you for taking the time to evaluate this manuscript. Your feedback is appreciated. Please find attached the document with our answers to your comments / suggestions and the revised manuscript.
We have included in the paper quantitative chemical determination for 3 types of active principles: flavones, PCA’s, and total polyphenols for all the medicinal plants from our culture. For a better quantification we also added the volumes of volatile oil obtained after distillation at Neo Clevenger Apparatus.
Reviewer 3 Report
This work was done to the initiation of a phytosociological study on certain types of medicinal plants:
The following types of medicinal plants:
Crop 1- Mentha x piperita L. and Melissa officinalis L. (aromatic medicinal plants from Lamiaceae family);
Crop 2 - Thymus vulgaris L. and Calendula officinalis L. (association in gastrointesti- nal diseases);
Crop 3 - Rosmarinus officinalis L. and Matricaria chamomilla L. (source of vola- tile oil);
Crop 4 - Hypericum perforatum L. and Chelidonium majus L. (associated in hepato- biliary disorders).
The Authors found that in all common crops compared to the control ones, the amount of vegetable product provided was much higher (for example - the thyme-rosemary crop produced 730 g of a fresh vegetable plant, compared with 540 g in the control crop; St. John's wort in culture with lemon balm delivered 1934 g of vegetable product, compared with 1423 g obtained from the control crop; mint has grown with lemon balm and produced a double amount of vegetable mass). Medicinal plants have developed both vertically and horizontally, and internal changes in their anatomical structure support the existence of a higher concentration of active principles. In consequence, the multitude of glandular hairs observed on a mint leaf, lemon balm leaf, rosemary leaf, and thyme herbs, denotes an increased concentration of volatile oil and the presence of an abundance of cells with carotenoid pigments was found in marigold flowers. Also, it can be stated that the dominant blackish punctuation on the leaves and petals of St. John's wort flowers, certifies the presence of secretory bags with hypericin in an increased concentration.
The idea of the search is great.
Planning and displaying the results in a simple and beautiful way.
My comments to the Authors:
Title: Please delete (a) from (a certain types)
Abstract section:
It has written very well.
Introduction section:
-P1 L36 (first paragraph in the introduction): Authors should add some citations, you are writing a scientific paper.
-P2 L47 (third paragraph in the introduction): add some citations.
- P2 L54: The end of paragraph I suggest adding this citation:
Ali, Muhammad Moaaz, et al. "Effect of Environmental Factors on Growth and Development of Fruits." Tropical Plant Biology (2021): 1-13.
Methods section:
-The authors explained that they planted two plants together in the introduction, but they did not explain that in the materials and methods, and how they planted the control crop (singly or what?? Page 2 Line 78-79).
-Please add the numbering for the sub-headings in the materials and methods (P2 L81 and P3L101).
Results section:
-P3L115: Authors have mentioned to their previous published results [6], I think will be better if you mention in discussion section.
-In table 1 and table 2 please change (Romanian în to English in).
-I think Figure 5 is a repetition of Figure 4, please if I understood right delete Figure 5.
-What is the y-axis unit in Figure 6 and Figure 11?
-In caption of Figure 4: Please change (Mentha piperita L. and Melissa officinalis L) to italic (Mentha piperita L. and Melissa officinalis L).
-In caption of Figure 7: Please change (Thymus vulgaris and Calendula officinalis) to italic (Thymus vulgaris and Calendula officinalis).
-In caption of Figure 8: Please change (Rosmarinus officinalis and Matricaria chamomilla & Rosmarinus officinalis and Thymus vulgaris) to italic (Rosmarinus officinalis and Matricaria chamomilla & Rosmarinus officinalis and Thymus vulgaris).
-P9L175: Please change (Melissa officinalis) to italic (Melissa officinalis).
-In caption of Figure 9: Please change (Hypericum perforatum and Chelidonium majus & Hypericum perforatum and Melissa officinalis) to italic (Hypericum perforatum and Chelidonium majus & Hypericum perforatum and Melissa officinalis).
-Will be better if authors use type of Palatino Linotype in Figures (6, 11, 17, and 18).
-In Figure 12, 13, 14, 15, and 16: please add the scale if possible. Some images have the scale, some are incomplete, and some do not have the scale.
- The numbers in Figure 17 and 18 are overlapping with each other. Please reduce the font and choose the type of Palatino Linotype.
-In table 1 some numbers are bold and some of numbers are not bold please check it carefully.
Discussion section:
-P16 L239 (two sentences): Authors should add some citations.
-P16 L249: this sentence (In addition to the issues studied, it has been reported that the practical restoration of degraded land is usually disrupted due to lack of basic information.); Authors should add some citations.
- P16L255: Change (Terminalia chebula Retz) to (Terminalia chebula Retz); last part should be not italic.
-P16 L259: Please delete (Mishra et al. 1997).
-P16 L262: All these sentences need citations (Research has also been carried out in our country, but in a rather small number. Such 262 a study took place in the Gurghiului Mountains, Romania, and medicinal and aromatic 263 plants from mountain hay and meadows were identified, being grouped according to the 264 dominant active principles used in phytotherapy. Two plant associations have been iden-265 tified: Festuco rubrae-Agrostietum capillaris Horvat 1951 and Poo-Trisetetum flavescentis 266 Knapp ex Oberdorfer 1957.).
- The discussion needs more depth to explain the mechanism of the relationship between plants that grow together.
References section: Please follow the journal style and there are some data missing from the references
Here missing data for some references:
- Ref. 2: Dengler, J. (2016). Phytosociology. International Encyclopedia of Geography: People, the Earth, Environment and Technology: People, the Earth, Environment and Technology, 1-6.
- Ref. 3: Hao, D. C., & Xiao, P. G. (2015). Genomics and evolution in traditional medicinal plants: road to a healthier life. Evolutionary Bioinformatics, 11, EBO-S31326.
- Ref. 6: LUȚĂ, E. A., Ghica, M., Costea, T., & ELENA, C. (2020). Phytosociological study and its influence on the biosynthesis of active compounds of two medicinal plants Mentha piperita L. and Melissa officinalis L. , Farmacia, 68(5), 919-924.
- Ref. 9: Sarkar, A. K., Dey, M., & Mazumder, M. (2017). Ecological status of medicinal plants of Chalsa forest range under Jalpaiguri division, West Bengal, India. International Journal of Herbal Medicine, 5(5), 196-215.
-Reference 2 and Reference 10 is same; please delete Reference 10 from the text and references.
Ref. 13: Samanpreet, S., Kamal, S., & Dushyant, S. (2019). Phytosociological studies on natural populations of Terminalia chebula Retz. in district Hamirpur, Himachal Pradesh. International Journal of Economic Plants, 6(4), 191-195.
I hope my comments improve the quality of your manuscript
Best regards

Author Response
Dear reviewer, thank you for taking the time to evaluate this manuscript. Your feedback is appreciated. Please find attached the document with our answers to your comments / suggestions and the revised manuscript.
My comments to the Authors:
Title: Please delete (a) from (a certain types)
Thank you for the suggestion. We made the change.
Abstract section:
It has written very well.
Thank you.
Introduction section:
-P1 L36 (first paragraph in the introduction): Authors should add some citations, you are writing a scientific paper.
-P2 L47 (third paragraph in the introduction): add some citations.
- P2 L54: The end of paragraph I suggest adding this citation:
Ali, Muhammad Moaaz, et al. "Effect of Environmental Factors on Growth and Development of Fruits." Tropical Plant Biology (2021): 1-13.
Thank you. We used this citation.
Methods section:
-The authors explained that they planted two plants together in the introduction, but they did not explain that in the materials and methods, and how they planted the control crop (singly or what?? Page 2 Line 78-79).
Thank you for your comment. We added the explanation.
-Please add the numbering for the sub-headings in the materials and methods (P2 L81 and P3L101).
Results section:
-P3L115: Authors have mentioned to their previous published results [6], I think will be better if you mention in discussion section.
-In table 1 and table 2 please change (Romanian în to English in).
-I think Figure 5 is a repetition of Figure 4, please if I understood right delete Figure 5.
-What is the y-axis unit in Figure 6 and Figure 11?
Y-axis represent plant heights shown in pictures in Figure 2 and Figure 4. We did the changes in this revised manuscript.
-In caption of Figure 4: Please change (Mentha piperita L. and Melissa officinalis L) to italic (Mentha piperita L. and Melissa officinalis L).
-In caption of Figure 7: Please change (Thymus vulgaris and Calendula officinalis) to italic (Thymus vulgaris and Calendula officinalis).
-In caption of Figure 8: Please change (Rosmarinus officinalis and Matricaria chamomilla & Rosmarinus officinalis and Thymus vulgaris) to italic (Rosmarinus officinalis and Matricaria chamomilla & Rosmarinus officinalis and Thymus vulgaris).
-P9L175: Please change (Melissa officinalis) to italic (Melissa officinalis).
-In caption of Figure 9: Please change (Hypericum perforatum and Chelidonium majus & Hypericum perforatum and Melissa officinalis) to italic (Hypericum perforatum and Chelidonium majus & Hypericum perforatum and Melissa officinalis).
Thank you for the suggestion. We made the changes for all of them to italic.
-Will be better if authors use type of Palatino Linotype in Figures (6, 11, 17, and 18).
Thank you. We chamged them according to the format.
-In Figure 12, 13, 14, 15, and 16: please add the scale if possible. Some images have the scale, some are incomplete, and some do not have the scale.
Thank you for the comment. We completed the scale where was not visible.
- The numbers in Figure 17 and 18 are overlapping with each other. Please reduce the font and choose the type of Palatino Linotype.
Thank you for the comment. We did the changes.
-In table 1 some numbers are bold and some of numbers are not bold please check it carefully.
Discussion section:
-P16 L239 (two sentences): Authors should add some citations.
-P16 L249: this sentence (In addition to the issues studied, it has been reported that the practical restoration of degraded land is usually disrupted due to lack of basic information.); Authors should add some citations.
- P16L255: Change (Terminalia chebula Retz) to (Terminalia chebula Retz); last part should be not italic.
-P16 L259: Please delete (Mishra et al. 1997).
Thank you for the comment. We did the changes.
-P16 L262: All these sentences need citations (Research has also been carried out in our country, but in a rather small number. Such 262 a study took place in the Gurghiului Mountains, Romania, and medicinal and aromatic 263 plants from mountain hay and meadows were identified, being grouped according to the 264 dominant active principles used in phytotherapy. Two plant associations have been iden-265 tified: Festuco rubrae-Agrostietum capillaris Horvat 1951 and Poo-Trisetetum flavescentis 266 Knapp ex Oberdorfer 1957.).
Thank you for the comment. We did the changes.
- The discussion needs more depth to explain the mechanism of the relationship between plants that grow together.
References section: Please follow the journal style and there are some data missing from the references
Here missing data for some references:
- Ref. 2: Dengler, J. (2016). Phytosociology. International Encyclopedia of Geography: People, the Earth, Environment and Technology: People, the Earth, Environment and Technology, 1-6.
- Ref. 3: Hao, D. C., & Xiao, P. G. (2015). Genomics and evolution in traditional medicinal plants: road to a healthier life. Evolutionary Bioinformatics, 11, EBO-S31326.
- Ref. 6: LUȚĂ, E. A., Ghica, M., Costea, T., & ELENA, C. (2020). Phytosociological study and its influence on the biosynthesis of active compounds of two medicinal plants Mentha piperita L. and Melissa officinalis L. , Farmacia, 68(5), 919-924.
- Ref. 9: Sarkar, A. K., Dey, M., & Mazumder, M. (2017). Ecological status of medicinal plants of Chalsa forest range under Jalpaiguri division, West Bengal, India. International Journal of Herbal Medicine, 5(5), 196-215.
-Reference 2 and Reference 10 is same; please delete Reference 10 from the text and references.
Ref. 13: Samanpreet, S., Kamal, S., & Dushyant, S. (2019). Phytosociological studies on natural populations of Terminalia chebula Retz. in district Hamirpur, Himachal Pradesh. International Journal of Economic Plants, 6(4), 191-195.
Thank you for the comment. We did the changes and reformulated the ideas.
Round 2
Reviewer 2 Report
Authors have significantly improved the manuscript but it still need some improvements
1- Add concluding remarks at the end of the abstract
2- I could not find any statistics on the data or statistical details in methodology. Adding the statistics is very important. Therefore, authors are strongly recommended to add the statistics to the data.
3- Statistics should contain at least (but not limited to) means and standard error/deviation
Author Response
Thank you very much for your review report. We did the changes and added the statistical analysis of the data. Please find attached the revised manuscript.
